

# The Dynamical Core of the Aeolus Statistical-Dynamical Atmosphere Model: Validation and Parameter Optimization

Sonja Molnos[1,2], Alexey V. Eliseev[1,3,4,5], Stefan Petri[1], Michael Flechsig[1], Levke Caesar[1,2],

Vladimir Petoukhov[1] and Dim Coumou[1,6]

[1] Potsdam Institute for Climate Impact Research, Potsdam, Germany

[2] Department of Physics, Potsdam University, Potsdam, Germany

[3] A.M. Obukhov Institute of Atmospheric Physics RAS, Moscow, Russia

[4] Kazan Federal University, Kazan, Russia

[5] Institute of Applied Physics, Russian Academy of Sciences, Nizhny Novgorod, Russia

[6] Institute for Environmental Studies (IVM), VU University Amsterdam

*Correspondence to*: S. Molnos (molnos@pik-potsdam.de)

**Abstract.** We present and validate a set of equations for representing the atmosphere's large-scale general

circulation in an Earth system model of intermediate complexity (EMIC). These dynamical equations have been implemented in *Aeolus*, which is a Statistical Dynamical Atmosphere Model (SDAM) and includes radiative transfer and cloud modules (Coumou, 2011; Eliseev, 2013). The statistical dynamical approach is computationally efficient, and thus enables us to do climate simulations at multi-millennia timescales, which is a prime aim of our model development. Further, this computational efficiency enables us to scan large and high-

dimensional parameter space to tune the model parameters.

We optimize the dynamical core parameter values by tuning all relevant dynamical variables to ERA-Interim reanalysis data (1983-2009) using monthly mean data of climatology data as well as the data for the El Niño and La Niña composites. We use a Simulated Annealing optimization algorithm, which approximates the global minimum of a high-dimensional function.

With non-tuned parameter values, the model performs reasonably in terms of its representation of zonal-mean circulation, planetary waves and storm tracks. The Simulated Annealing optimization improves in particular the model's representation of the northern hemisphere jet stream and storm tracks as well as the Hadley circulation.

The regions of high azonal wind velocities (planetary waves) are accurately captured for all validation experiments. The zonal-mean zonal wind and the integrated lower troposphere mass flux show good results in

particular in the Northern Hemisphere. In the Southern Hemisphere, the model tends to produce too weak zonal-mean zonal winds and a too narrow Hadley circulation. We discuss possible reasons for these model biases as well as planned future model applications.





**Keywords:**

Earth system model of intermediate complexity, Statistical- Dynamical Atmosphere Model, Optimization, model parameter tuning

## 1    Introduction

Numerical models of the Earth system play a key role in our understanding of physical processes in Earth and Atmosphere and can be used to simulate past and future climate changes.

General circulation models (GCMs) are physically the most realistic tools for studying and modelling climate variability and climate change in the Earth system. However, due to their relatively high-resolution, they are costly in terms of CPU runtime limiting their applicability to study climate variability over long (~millennia) timescales.

On the other hand, highly idealized and computational efficient models for the climate system are able to simulate long time periods, but those are often box-, one- or two dimensional models describing only a limited number of processes or feedbacks of the real world. Hence their application is limited, but they have been applied to study paleo climate (Berger et al., 1992; Harvey, 1989) and future global change (Xiao et al., 1997).

A third class of models are so-called intermediate complexity climate models (EMICs) which form a compromise between the computationally expensive (but more realistic) GCMs and the very simplified models (Claussen et al., 2002). The number of processes and feedbacks are comparable to GCMs, however due to a reduction in resolution and/or complexity of some model components, it is possible to study climate simulations up to multi-millennia timescales (Eliseev et al., 2014a, 2014b; Ganopolski et al., 2001; Montoya et al., 2005). Other applications include determining quick assessments of climate change impacts or run thousands of parameter sensitivity experiments (Knutti et al., 2002; Schmittner and Stocker, 1999).

EMICs are thus particularly useful for understanding the different roles of different Earth components on very long timescales (multi-millennia and longer) and consequently form useful tools complementary to GCMs. Internal climate processes on such very long timescales are primarily driven by ocean and ice dynamics (Holland et al., 2001; Latif, 1998; Polyakov et al., 2003), with the atmosphere's role likely limited to globally distributing any perturbations to the system. In GCMs, it is however often the atmosphere which takes most of the computational load due to the need to resolve synoptic weather systems, which requires a high-resolution discretization in space and time. For these reasons, a key step in development of EMICs intended for studying ocean and ice dynamics on multi-millennial timescales, is the derivation and validation of statistical-dynamical equations which accurately represent atmosphere dynamics (Coumou et al., 2011).

In section 2 we present the equations of the Aeolus dynamical core with the derivation of these equations presented in the Suppl. Mat. . The dynamical core is coupled with a convective plus 3-layer stratiform cloud scheme developed by Eliseev et al. (2013). In section 3 we describe the experiment setup and the used reanalysis data sets. In section 4 we explain the model discretization and in section 5 we introduce our specific calibration method. For parameter optimization of the wind velocities we use *Simulated Annealing* which approximates the global minimum of a high-dimensional function (Flechsig et al., 2013). In section 6, we present Aeolus' dynamical fields with pre-optimized and optimized parameters and compare them with observations and output



from models of the Coupled Climate Modeling Experiment Phase 5 (CMIP5). We conclude by discussing performance and limitations of the model in section 7.

## 5    2    Governing Equations

### 2.1    General structure of the atmosphere

*Aeolus* is a 2.5- dimensional statistical-dynamical model, with the vertical dimension largely parameterized and only coarsely resolved and it therefore belongs to the class of intermediate complexity atmosphere models. Water and energy conservations is achieved via a set of 2-dimensional, vertically averaged prognostic equations
for temperature and water vapor (Petoukhov et al., 2000).

The 3-dimensional structure is described by these 2-dimensional fields with the vertical dimensions reconstructed using an equation for the lapse rate and assuming an exponential profile for specific humidity

The equations of the dynamical core of *Aeolus* describe the time evolution of synoptic-scale transient waves (or storm tracks), quasi-stationary planetary waves and the zonal-mean wind. Thus, following classical statistical-
dynamical approaches (Dobrovolski, 2000; Imkeller and von Storch, 2012), the key assumption is that the wind velocity field $V$ can be split into a synoptic scale ($V'$) component (2-6 days period) and a large-scale long-term component ($\langle V \rangle$) (Fraedrich and Böttger, 1978) such that

$$V = \langle V \rangle + V' = \{\langle u \rangle, \langle v \rangle, \langle w \rangle\} + \{u', v', w'\} \qquad (1)$$

The variables $u$, $v$ and $w$ describe the wind velocity in zonal, meridional and vertical direction. The brackets $\langle ... \rangle$ symbolize time averaged quantities and the prime ($'$) indicates deviations from this time averaged field. The
large-scale long term $\langle V \rangle$ is subdivided into a zonal-mean $\overline{\langle V \rangle}$ and an azonal component $\langle V^* \rangle$ :

$$\langle V \rangle = \overline{\langle V \rangle} + \langle V^* \rangle \qquad (2)$$

The large scale, zonal-mean zonal wind velocity $\overline{\langle u(z, \phi) \rangle}$ with height above surface $z$ and latitude $\phi$ is estimated using the general geostrophic wind equation:

$$\overline{\langle u(z, \phi) \rangle} = -\frac{1}{f}\left(\frac{1}{\rho_0}\langle\overline{\frac{\partial p_0}{\partial \phi}}\rangle + \int_0^z \frac{g}{T_0}\langle\overline{\frac{\partial T}{\partial \phi}}\rangle\, dz\right), \qquad (3)$$

whereby the sea level pressure gradient is calculated by $\langle\overline{\frac{\partial p_0}{\partial \phi}}\rangle = \frac{v^*\rho|f|}{-c_\alpha \sin \alpha}$ and $\alpha$ is the cross-isobar angle defined as in Coumou et al. (2011). The variable $v^*$ is the azonal meridional wind velocity, $\rho$ is air density, f, the
Coriolis parameter, reference air density $\rho_0$ and $\phi$ is the latitude. The Coriolis parameter $f$, reference air densitiy $\rho_0$, reference temperature $T_0$ and gravitational acceleration $g$ (See Petoukhov et al., 2000).

As derived in S1.2, the large scale, zonal-mean meridional wind velocity $\overline{\langle v(z, \phi) \rangle}$ is given by



$$\overline{\langle v(z,\phi)\rangle} \tag{4}$$

$$= \frac{d1 * \left(-2\tan(\phi)\left(\overline{\langle u^*v^*\rangle} + \overline{\langle u'v'\rangle}\right)\right) + d2 * \left(\frac{\partial}{\partial\phi}\left(\overline{\langle u^*v^*\rangle} + \overline{\langle u'v'\rangle}\right)\right) + d3 * \left(\left(\frac{z}{H_0} - 1\right)\frac{\partial\overline{\langle u\rangle}}{\partial z}a\right) + d4 * (A)}{n1 * \left(\tan(\phi)\overline{\langle u\rangle}\right) + n2 * \left(-\frac{\partial\overline{\langle u\rangle}}{\partial\phi}\right) + n3 * (2\Omega a\sin(\phi))},$$

Where $d_1, d_2, d_3, d_4, n_1, n_2$ and $n_3$ are tunable parameters and

$$A = \frac{\mathcal{L}\overline{\langle P_{co}\rangle}}{H_0}\frac{\overline{\langle u_{sf}\rangle}}{\Gamma_a - \Gamma_0 - \Gamma_1(T_a - T_0)\left(1 - a_q q_s^2\right) + \Gamma_2 n_c}$$

The vertical friction coefficient $K_z$, atmosphere scale height $H_0$, and Earth's angular velocity $\Omega$ as well as dry

adiabatic lapse rate $\Gamma_a$, latent heat of evaporation $\mathcal{L}$ and model parameters $\Gamma_0, \Gamma_1, \Gamma_2, a_q, T_0$ are explained in **Table 1**. $T_a$ is a temperature which would occur near the surface if the lapse rate did not change within the planetary boundary layer (PBL), $q_s$ is the surface air specific humidity and $n_c$ is the cumulus cloud amount. The variable $\overline{\langle P_{co}\rangle}$ is the convective precipitation rate and is calculated by the cloud model (Eliseev et al., 2013). The variable $\overline{\langle u_{sf}\rangle}$ is the surface wind and stated in the Supl. Ment. .

The azonal component of the wind field describes quasi-stationary planetary waves and depends on the latitude, longitude and height. At the equivalent barotropic level (EBL), azonal geostrophic components of horizontal velocities are computed employing the definition of the stream function $\psi$ depending on latitude $\phi$ and longitude $\lambda$,

$$\langle u^*_{EBL}(\lambda,\phi)\rangle = -\nabla_\phi\langle\psi^*_{EBL}\rangle \tag{5}$$

$$\langle v^*_{EBL}(\lambda,\phi)\rangle = \nabla_\lambda\langle\psi^*_{EBL}\rangle \tag{6}$$

whereby the stream function can be subdivided into contributions from thermally and orographically induced

waves depicted by subscript *th* and *or* respectively. They are considered to be additive due to linearity of the barotropic vorticity equations such that

$$\langle\psi^*_{EBL}\rangle = \Psi_0 \cdot \langle\psi^*_{th,EBL}\rangle + \langle\psi^*_{or,EBL}\rangle \tag{7}$$

The parameter $\Psi_0$ is a tuning parameter which is necessary since smoothing is applied to dampen local moisture feedbacks in the model. This smoothing however reduces spatial gradients in $\psi^*_{EBL}$ and therefore $u^*_{EBL}$ and $v^*$ itself.

The zeroth order solution of the thermally induced waves of the barotropic vorticity equation is given by (see Suppl. Information S.2.):





$$\langle \psi^*_{th,0,EBL} \rangle = -\langle T^*_{EBL} \rangle \frac{ag}{\Omega \rho_0 T_0^2 \cos\phi} \nabla_\phi \int_0^{z_{EBL}} \rho_0 \langle [T(z)] \rangle \, dz \qquad (8)$$

It is solved at two beta-planes, for the northern and southern hemisphere, respectively:

$$\langle \psi^*_{th,0,EBL} \rangle_{NH} = -\langle T^*_{EBL} \rangle \frac{ag}{\Omega \rho_0 \, T_0^2 \cos\beta_{NH}} \nabla_\phi \int_0^{z_{EBL}} \rho_0 \langle [T(z)] \rangle \, dz \qquad (9)$$

$$\langle \psi^*_{th,0,EBL} \rangle_{SH} = -\langle T^*_{EBL} \rangle \frac{ag}{\Omega \rho_0 \, T_0^2 \cos\beta_{SH}} \nabla_\phi \int_0^{z_{EBL}} \rho_0 \langle [T(z)] \rangle \, dz \qquad (10)$$

The beta-plane is an approximation, in which the Coriolis parameter is linearized to a reference latitude, respectively $\beta_{NH}$ and $\beta_{SH}$ for the Northern Hemisphere and Southern Hemisphere. In the tropical belt the variable $\langle \psi^*_{th,0,EBL} \rangle$ is interpolated linearly between the beta-planes

The integrated heat content in equation (8) ($I_v = \int_0^{z_{EBL}} \rho_0 \langle [T^*(z)] \rangle \, dz$) is calculated analytically by assuming a constant lapse rate $\Gamma$ such that $T(z) = T(z_{EBL}) - \Gamma(z - z_{EBL})$. One obtains

$$I_v = \rho_0 ([T(z_{EBL})] - [\Gamma] z_{EBL}) H_0 \left(1 - e^{-\frac{z_{EBL}}{H_0}}\right) - \Gamma \rho_0 H_0 \left\{ (H_0 - z_{EBL}) \left(e^{-\frac{z_{EBL}}{H_0}} - 1\right) \right\}$$

In addition, $I_v$ is smoothed by 5 points in latitude to avoid numerical artefacts which may arise due to spatial differentiating.

To remove possible singularities near the poles, at high latitudes the stream function is dampened by a fourth
order interpolation function. Planetary waves at other tropospheric level are directly calculated from those at the EBL (see in the Suppl. Mat.) .

The time averaged kinetic energy of transient eddies $\langle E'_k \rangle = \frac{1}{2}(\langle u'^2 \rangle + \langle v'^2 \rangle)$ is determined using the statistical-dynamical equations as described in Coumou et al., (2011):

$$\frac{\partial \langle E'_k \rangle}{\partial t} = -\langle \boldsymbol{V} \rangle \cdot \nabla \langle E'_k \rangle + \langle u' \boldsymbol{V}' \rangle \cdot \nabla \langle u \rangle - \langle v' \boldsymbol{V}' \rangle \cdot \nabla \langle v \rangle + K_{fh} \nabla_H \langle E'_k \rangle - K_{fs} \langle E'_k \rangle + f \big( \langle u' v'_{ag} \rangle - \langle v' u'_{ag} \rangle \big) \qquad (11)$$

Here, $K_{fh}$ and $K_{fz}$ are internal atmospheric small/meso-scale friction coefficients in the horizontal and vertical direction respectively, $K_{fs}$ is the surface friction coefficient, $f$ the Coriolis parameter and subscript "ag" denotes ageostrophic terms.

By assuming that the vertical (baroclinic) flux term is equipportioned between the zonal and meridional kinetic
energy component, we can split the Eq. (9) into three separate equations for $\langle u'^2 \rangle, \langle v'^2 \rangle$ and $\langle u'v' \rangle$:





$$\frac{\partial \langle u'^2 \rangle}{\partial t} = -\langle \boldsymbol{V} \rangle \cdot \nabla \langle u'^2 \rangle - 2\langle u'^2 \rangle \frac{\partial \langle u \rangle}{\partial x} - 2\langle u'v' \rangle \frac{\partial \langle u \rangle}{\partial y} + K_{syn} \left[ \left( \frac{\partial \langle u \rangle}{\partial z} \right)^2 + \left( \frac{\partial \langle v \rangle}{\partial z} \right)^2 \right]$$
$$+ K_{fh}\Delta_H \langle u'^2 \rangle + K_{fz}\Delta_z \langle u'^2 \rangle - K_{fs}\langle u'^2 \rangle + f\left( \langle u'v'_{ag} \rangle - \langle v'u'_{ag} \rangle \right) \tag{12}$$

$$\frac{\partial \langle v'^2 \rangle}{\partial t} = -\langle \boldsymbol{V} \rangle \cdot \nabla \langle v'^2 \rangle - 2\langle v'^2 \rangle \frac{\partial \langle v \rangle}{\partial y} - 2\langle u'v' \rangle \frac{\partial \langle v \rangle}{\partial x} + K_{syn} \left[ \left( \frac{\partial \langle u \rangle}{\partial z} \right)^2 + \left( \frac{\partial \langle v \rangle}{\partial z} \right)^2 \right] + K_{fh}\Delta_H \langle v'^2 \rangle +$$
$$K_{fz}\Delta_z \langle v'^2 \rangle - K_{fs}\langle u'^2 \rangle + f\left( \langle u'v'_{ag} \rangle - \langle v'u'_{ag} \rangle \right) \tag{13}$$

$$\frac{\partial \langle u'v' \rangle}{\partial t} = -\langle \boldsymbol{V} \rangle \cdot \nabla \langle u'v' \rangle - \langle u'\boldsymbol{V} \rangle \cdot \nabla \langle v \rangle - \langle v'\boldsymbol{V} \rangle \cdot \nabla \langle u \rangle \; + K_{fh}\Delta_H \langle u'v' \rangle + K_{fz}\Delta_z \langle u'v' \rangle -$$
$$K_{fs}\langle u'v' \rangle + f\left( \langle u'u'_{ag} \rangle - \langle v'uv'_{ag} \rangle \right) \tag{14}$$

The parameterizations for $K_{syn}$ and $f\left( \langle u'u'_{ag} \rangle - \langle v'uv'_{ag} \rangle \right)$ were found and derived in Coumou et al. (2011).

This provides us with a coupled set of equations for $\overline{\langle u \rangle}, \overline{\langle v \rangle}, \langle u^* \rangle, \langle v^* \rangle, \langle u'^2 \rangle, \langle v'^2 \rangle$ and $\langle u'v' \rangle$, which can be solved. Cross terms like $\overline{\langle u^*v^* \rangle}$ can be determined by multiplying $\langle u^* \rangle$ with $\langle v^* \rangle$ and taking the zonal-mean of that quantity. All derivatives are determined numerically. The values of the parameters are listed in **Table 1**.

### 3   Forcing data and Reanalysis data sets

The simulations were forced by multi-year averages of monthly mean climatological, El Niño and La Niña months data (surface temperature, specific humidity, temperature at 500 mb, geopotential height at 500 mb and 1000 mb) using ERA-Interim Reanalysis (Dee et al., 2011) for 1983-2009 as our aim is that *Aeolus* captures year-to-year variability associated with the ENSO cycle. We identified  87 El Niño ( 74 La Niña ) months using 3 month running mean of ERSST.v4 SST anomalies (Huang et al., 2016) and applying the definition that those

15   months, where at least 5 consecutive overlapping seasons of SST anomalies are greater than 0.5K ( less than -0.5K), are El Niño (La Niña) events.

Multi-year averages of monthly mean, El Niño and La Niña months cumulative cloud fraction is taken from ISSCP (Rossow and Commission, 1996). The spatial resolution is 2.5 × 2.5 degrees lat × lon and the time range is 1983-2009.

We chose this time period, because the cumulative cloud fraction data is only available for this time period, which is needed to calculate the lapse rate.

To avoid strong temperature gradients in the specified boundary conditions for the numerical experiments, we use the lapse rate equation to calculate temperatures at 1000 mb from those at 500 mb. We first calculate the





lapse rate using the temperature field and specific humidity using the equation as in Petoukhov et al. (2000) at 1000 mb. Then, we recalculate temperature field at 1000 mb by using the temperature field at 500 mb and the linear lapse rate equation. This way we ensure that the temperature at 500 mb is close to observations, and at the same time we have a vertical temperature profile realistic for a model like Aeolus. Since the ERA-Interim

500 mb temperatures contain an orographic component, we exclude $\langle \psi^*_{or,EBL} \rangle$ in equation (7) in order not to incorporate orographic forcing of planetary waves twice.

We optimized the numerical solutions of the wind velocities $u^*$, $v^*$ and $\overline{\langle u \rangle}$ as well as eddy kinetic energy $\langle E'_k \rangle$ at $p = 500$ mb. To compare the strength and position of the Hadley and Ferrel cells between observation and model, we calculate a zonal-mean mass flux $\overline{\langle m \rangle}$ in the lower troposphere using the zonal-mean meridional wind

velocity $\overline{\langle v \rangle}$ at levels between 1000 mb – 500 mb and assuming exponential decay of air density with height (Petoukhov et al., 2000).

For use with Aeolus, all data sets are interpolated to $3.75 \times 3.75$ degrees lat $\times$ lon spatial resolution.

### 4 Model discretization

Aeolus operates on a reduced grid to overcome the restriction of small time steps near the poles due to the CFL

criteria (Jablonowski et al., 2009). In the grid generation, longitudinally adjacent cells are merged, if their zonal width in meters would be less than half of the cell width at the equator.

This way the reduced grid has the same resolution as a regular grid at the equator, but, at nominal resolution $3.75 \times 3.75$ degrees lat $\times$ lon, around the poles only 6 cells are defined. On the regular grid, the maximum permissible time step due to the CFL criteria would be ca. 5 min, while the limit for the reduced grid is ca. 2

20  hours.

### 5 Calibration

The equations (1) – (14) are implemented in *Aeolus* and numerically solved on a $3.75 \times 3.75$ degrees lat $\times$ lon reduced grid with 5 tropospheric height levels (1000 m, 3000 m, 5000 m, 9000 m).

The calibration of the winds is divided into two parts. First, we optimize the dynamical variables primarily

driven by the thermal state of the atmosphere: The azonal velocities in zonal and meridional direction $\langle u^* \rangle$ and $\langle v^* \rangle$ as well as the zonal-mean zonal wind velocity $\overline{\langle u \rangle}$. In the second step, we tune the zonal-mean synoptic kinetic energy $\overline{\langle E'_k \rangle}$ and the lower troposphere integrated mass flux $\overline{\langle m \rangle}$, which solely depends on the zonal-mean meridional wind $\overline{\langle v \rangle}$.

A common approach for parameter tuning is *Simulated Annealing* (Ingber, 1996) It is one experiment type in the

multi-run simulation environment *SimEnv* for sensitivity and uncertainty analysis of model output (Flechsig et al., 2013) which we use for all calibration experiments.

For each model run, the thermal state of the atmosphere is kept constant (and initialized as described above) and the dynamical core is equilibrated to this thermal state. This typically requires only a few time steps. Since we tune only the parameters of the dynamical core, Aeolus first calculates the clouds using its cloud scheme



(Eliseev et al., 2013) to determine lapse rate and initialize the 3D thermal state. After that only the state of the dynamical core is updated.

### 5.1    Dynamical Core Tuning - Step 1

For a good starting point the parameters are first tuned manually providing "pre-optimised" values. Next, we define physically realistic parameter-ranges for automatic tuning as listed in **Table 2**.

For the azonal wind velocities we use a weighting function which excludes tropics (from 10°S to 10°N) and polar regions (poleward of 60° for the Southern Hemisphere and poleward 70° for the Northern Hemisphere) such that the mid-latitudes, where planetary waves are important, are optimized.

The non-excluded grid as well as the zonal-mean zonal wind is weighted by $w(\phi) = |\cos(\phi)|$.

The total skill score for the scheme in step 1 is calculated by multiplying the individual skills for the azonal velocities in zonal and meridional direction ($S_{u^*}, S_{v^*}$) and the skill for the zonal-mean zonal wind velocity($S_{\overline{\langle u \rangle}}$):

$$S = S_{u^*} S_{v^*} S_{\overline{\langle u \rangle}}$$

The goal of the optimization procedure is to maximize the skill S.

Skill score functions for individual variables are computed as in Taylor (2001)

$$S(\phi, \lambda, t) = \frac{(1 + r_X)^4}{\left(A_X + \frac{1}{A_X}\right)^2} \qquad (15)$$

In Eq. ( 15 ) $r_X$ is the coefficient of the spatial correlation between the area-weighted modelled and observed fields of $X$; and $A_X$ is the so-called relative spatial variation calculated according to

$$A_X = \frac{A_{X,M}}{A_{X,O}}. \qquad (16)$$

Here, the variable $A_{X,M}$ is the spatial average of $|X_M - X_{M,g}|$ and $X_{M,g}$ is a globally averaged value of the modelled field $X_M$. The observed field is similarly defined by $A_{X,O}$.

### 5.2    Dynamical Core Tuning - Step 2

For tuning the zonal-mean meridional wind velocity $\overline{\langle v \rangle}$ and in particular the strength and width of the Hadley cell we use the vertical integral of the lower tropospheric integrated mass flux $\overline{\langle m \rangle}$. In addition, we tune the zonal-mean area-weighted synoptic kinetic energy $\langle E_k' \rangle$. Both variables strongly depend on the dynamic fields tuned in step 1 which is the reason for tuning them in a separate second step.

Total skill score for the scheme in step 2 is calculated by multiplying the individual skills for the vertical integral of lower troposphere mass flux ($S_{\overline{\langle m \rangle}.}$) as well as the eddy kinetic energy ($S_{\langle E_k' \rangle}$)

$$S = S_{\overline{\langle m \rangle}.} S_{\langle E_k' \rangle}$$

The goal of the optimization procedure is again to maximize skill S.





The skill score function for the eddy kinetic energy is given by the Taylor skill score function, e.g. Eq. (15).

The skill score function for the mass flux consists of the product of the correlation of observation and model as well as the mean mass flux of the Hadley cell. The skill score is then calculated by

$$S_{\overline{\langle m \rangle}} = \left( \text{mean}_{\text{Hadley\_Obs}} - \text{mean}_{\text{Hadley\_Model}} \right)^2 r_X^2 \qquad (17)$$

Here $r_X$ is the coefficient of the spatial correlation between area-weighted modeled and observed fields (as in Eq. (15)), $\text{mean}_{\text{Hadley\_Model}}$ and $\text{mean}_{\text{Hadley\_Obs}}$ are the mean values of the area-weighted modeled and observed fields. We use this more elaborate skill function to promote a proper Hadley circulation in the model.

The weights of the lower troposphere mass flux $\overline{\langle m \rangle}$ are calculated according to:

$$w(\phi) = \begin{cases} |\cos(\phi)|, & \phi > 60°S \\ 0 & \phi \leq 60°S \end{cases} \quad .$$

For calculating the mean intensity of the Hadley cell we determine the roots of the mass flux in observation data close to 0° and 30° which determine the Hadley cell latitudinal boundaries. This way, we have 36 values for the boundaries of the northern Hadley cell. Between these latitudinal borders we calculate the mean strength of the Hadley cell.

In **Table 3** the manually tuned (or pre-optimized) parameters and their ranges are listed.

## 6    Results

### 6.1    Results of Calibration – Step 1

We compared the numerical solutions using the optimized parameters for the wind fields $\langle u^* \rangle$, $\langle v^* \rangle$ and $\overline{\langle u \rangle}$ of climatological monthly averages, El Niño and La Niña months from ERA-Interim Reanalysis (Dee et al., 2011) for 1983-2009.

The figures for azonal wind velocities are divided into 6 subplots: The left column shows observational data and the right column model data. The top row shows climatological monthly averages, the middle multi-year averages of El Niño months and the bottom row multi-year averages of La Niña months.

In **Figure 1** and **Figure 2** the azonal zonal wind velocity for February and August at 500 mb are displayed, respectively. The figures show that with optimized parameters the model reasonably reproduces the main observed features both in terms of spatial position and magnitude. In particular the extra-tropical planetary waves are well captured with some minor discrepancies in the tropics. Both the seasonal cycle and the response to the ENSO cycle are well captured by the model.

**Figure 3** and **Figure 4** show the same type of plots for $\langle v^* \rangle$. Also for the meridional wind velocity the most important features of the reanalysis data are well represented in the model. The model results coincide well in wind strength and spatial pattern with the reanalysis data. The wind strength in winter, for both climatological and El Niño months are stronger than for winter La Niña months. In summer the opposite is seen for both model and reanalysis data.



In **Figure 5** the zonal-mean zonal wind velocity $\overline{\langle u \rangle}$ at 500 mb is shown with the orange line representing reanalysis data, red representing model data with optimized parameters, and gray representing model data with pre-optimized parameters. The figure is subdivided into six subplots: The top row depicts $\overline{\langle u \rangle}$ in February and the bottom row shows $\overline{\langle u \rangle}$ in August and the columns providing respectively climatological data, El Niño data and La Niña data. It is noticeable that the results obtained with pre-optimized parameters are already reasonable but that optimization hardly improves model results. The Northern Hemisphere $\overline{\langle u \rangle}$ profile is well resolved in both seasons and for El Niño and La Niña months. Parameter optimization slightly improves results in the tropics. The modeled amplitude of $\overline{\langle u \rangle}$ in the Southern Hemisphere is too small in February for all plots, and in August too high.

The optimized parameters are listed in **Table 2**. The $\beta_{NH}$ in the Northern Hemisphere has a higher value, whereas the $\beta_{SH}$ in the Southern Hemisphere has a lower value.

The last parameter is $\Psi_0$ and is changed to a higher value in order to strengthen velocities in $\langle v^* \rangle$ and $\langle u^* \rangle$.

### 6.2 Results of Calibration – Step 2

We compared the numerical solutions using the optimized parameters for the zonal-mean lower troposphere integrated mass flux $\overline{\langle m \rangle}$ and eddy kinetic energy $\overline{\langle E'_K \rangle}$.

The plots in **Figure 6** show that in general the monthly mean zonal-mean mass flux calculated with optimized parameters matches better with observational data. The Hadley cell is generally too weak with pre-optimized parameters, which improves with the optimized parameters. The ENSO cycle is clearly visible and the width of the Hadley cell is wider compared to results with pre-optimized parameters. However, the width of the Hadley cell (especially in August) is still too small compared to the width of the Hadley cell obtained by reanalysis data. The figure shows only plots with a latitudinal range from 60°S to 90°N as reanalysis data is spikey over Antarctica.

**Figure 7** shows the zonal-mean eddy kinetic energy $\overline{\langle E'_K \rangle}$. We show the same color code as in **Figure 6.** Northern Hemisphere modeled $\overline{\langle E'_K \rangle}$ profile is again well resolved in both seasons and for El Niño and La Niña months with the parameter optimization. Smaller spikes vanish such that the modeled $\overline{\langle E'_K \rangle}$ better matches the observed data. However, the modeled optimized $\overline{\langle E'_K \rangle}$ curve in the Southern Hemisphere does not substantially improve compared to pre-optimized parameters.

In **Figure 8** and **Figure 9** the eddy kinetic energy $\overline{\langle E'_K \rangle}$ for February and August is displayed. The left column shows observational data and the right column model data. The top row presents climatological monthly averages, the middle El Niño months and the bottom row La Niña months.

The spatial position and the magnitude are well captured, seasons and the ENSO-cycles are also well resolved with some discrepancies in the tropics (i.e. the region over the Atlantic and Pacific Ocean) and the Southern Hemisphere. In February and August $\overline{\langle E'_K \rangle}$ is stronger for both climatology and El Niño in the Northern



Hemisphere than in the Southern Hemisphere. Only in La Niña months, $\overline{\langle E_K' \rangle}$ is weaker in the Northern Hemisphere.

The optimized parameters are listed in **Table 3**. The parameters $U0$ and $m$ for optimizing the eddy kinetic energy are greater than the manually tuned values.

The parameters $d3$ and $n3$ are close to one, whereas the parameters $d2, d4$ and $n1$ are close to 2 and have a strong impact on the amplitude of the Hadley cell and the Ferrell cell. The parameter with the smallest influence is $d1$ ($d1 = 0.41$).

### 6.3    Comparison to CMIP5 Models

**Figure 10** shows the comparison of February and August $\overline{\langle E_K' \rangle}, \overline{\langle u \rangle}$  and $\overline{\langle m \rangle}$ between CMIP5 (grey lines),
Aeolus (red) and ERA-Interim data (orange). In General CMIP5 models represent the $\overline{\langle E_K' \rangle}$ and $\overline{\langle u \rangle}$ very well in both Hemispheres. However, in the Southern Hemisphere, the storm tracks, i.e. $\overline{\langle E_K' \rangle}$, of all models are too weak compared to observations with Aeolus on the lower end of the CMIP5 range. Further, some individual CMIP5 models can have too low or too high $\overline{\langle E_K' \rangle}$ and $\overline{\langle u \rangle}$ as compared to ERA - Interim, similar to Aeolus.

The CMIP5 multi-model mean of $\overline{\langle m \rangle}$ appears to be close to the reanalysis and most models reproduce this well.
Still, some CMIP5 models can differ strongly from $\overline{\langle m \rangle}$ in ERA-Interim with some spikey behavior. Nevertheless, the width and strength of the Hadley cell is in most models well presented, but the Ferrell cell is often too strong. Aeolus results give reasonable strength and width of the Ferrell cell, but the width of the Southern Hemisphere Hadley cell in August is too small compared to both reanalysis and CMIP5 models.

### 7    Summary and Discussion

In this paper we presented the atmosphere model *Aeolus*, which is a statistical-dynamical atmosphere model and belongs to the class of intermediate complexity models. The equations of *Aeolus* are time-averaged and the model has a spatial resolution of $3.75° \times 3.75°$. The 3-dimensional structure of *Aeolus*  is reconstructed using a set of 2-dimensional, vertically averaged prognostic equations for temperature and water vapor (Petoukhov et al.,
2000).  The advantage of such type of models is the fast computation time and for that reason the possibility to study and simulate long time periods as well as conduct sensitivity experiments.

We performed parameter optimization of the dynamical core consisting of a large multi-dimensional parameter space, which is in a high parameter range and can be searched due to its fast computation time. For this approach we used the optimization algorithm *Simulated Annealing*, which approximates the global minimum of a high-
30 dimensional function. We divided the calibration into two parts. At first, the azonal velocities in zonal and meridional direction as well as the zonal-mean zonal wind velocity were optimized, because they are primarily driven by the thermal state of the atmospheres. In the second step we optimized the zonal-mean synoptic kinetic energy and the lower troposphere integrated mass flux, and hence the zonal-mean meridional velocity, since those variables depend strongly on variables of step 1.

The results of the winds and eddy kinetic energy are in reasonable agreement with the reanalysis data and



showed that our model is able to reproduce the dynamic response from the season-cycle as well as the ENSO cycle which is a prime goal of our model development efforts. Parameter optimization in particular improves representation of the Hadley cell in terms of strength and width.

In the Southern Hemisphere the dynamical fields tend to be too weak. This model bias might be related to the missing Antarctica ice sheet, upper-tropospheric ozone, the constant lapse rate assumption, or fundamental limitations of the equations. These possibilities will be analysed in future work using the coupled Potsdam Earth Model (POEM) to which Aeolus has been coupled.

Compared to CMIP5 models, Aeolus reasonable well captures the dynamical state of the atmosphere in the Northern Hemisphere, particularly for monthly mean eddy kinetic energy $\overline{\langle E_K' \rangle}$, zonal-mean wind velocity $\overline{\langle u \rangle}$ and mass flux $\overline{\langle m \rangle}$. Especially the mass flux of the Ferrell cell is better captured than in other models, whereas the Southern Hemisphere Hadley cell width of Aeolus in August is too small compared to CMIP5 models.

## 8 Code and data availability

Code and data are stored in PIK's long term archive, and are made available to interested parties on request

## 9 Team list

S. Molnos, A. V. Eliseev, S. Petri, M. Flechsig, L. Caesar, V. Petoukhov and D. Coumou

## 10 Competing interests

The authors declare that they have no conflict of interest

## 11 Acknowledgements

We thank ECMWF for making the ERA-Interim available.

The work was supported by the German Federal Ministry of Education and Research, grant no. 01LN1304A, (S.M., D.C.).

A.V.E. contribution was partly supported by supported by the Government of the Russian Federation (agreement No. 14.Z50.31.0033).

The authors gratefully acknowledge the European Regional Development Fund (ERDF), the German Federal Ministry of Education and Research and the Land Brandenburg for supporting this project by providing resources on the high performance computer system at the Potsdam Institute for Climate Impact Research.

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

**Figures**




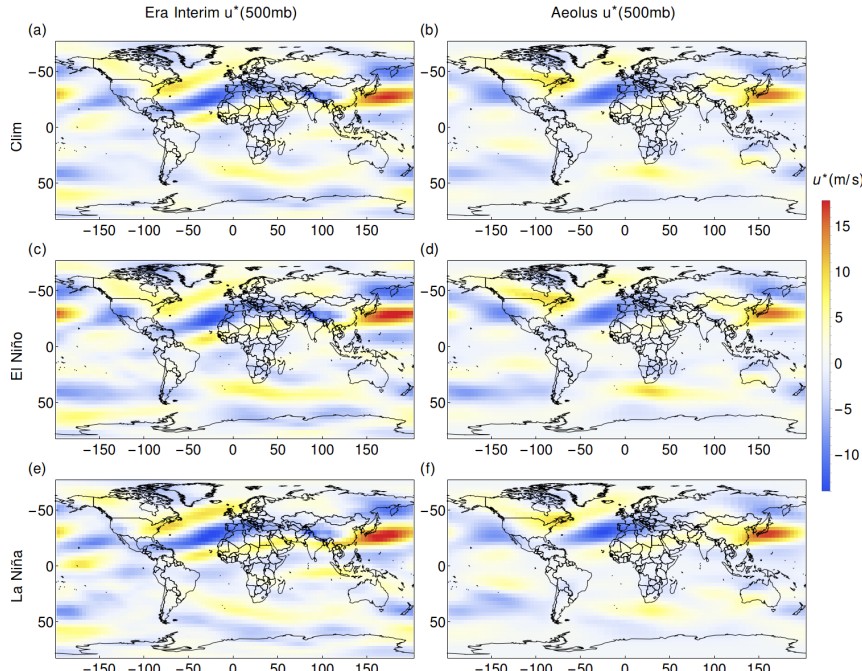

**Figure 1 Monthly mean azonal large scale zonal wind u\* in February at 500mb. The first column shows the results from observation data and second column shows the results from Aeolus received by optimized parameters.**



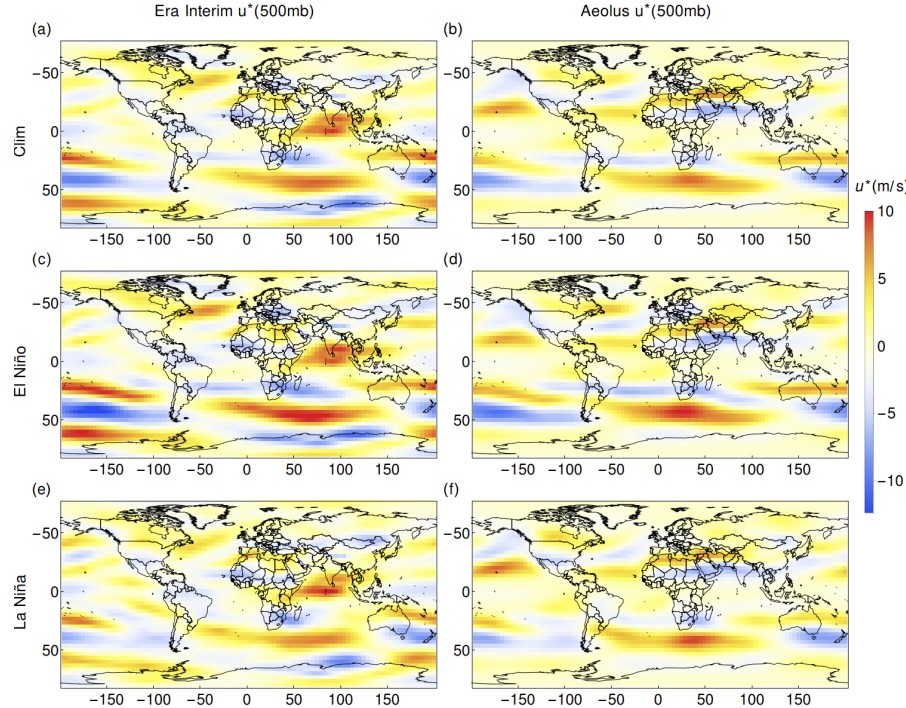

**Figure 2 Monthly mean azonal large scale zonal wind velocity u\* in August at 500mb (compare Fig. 1).**





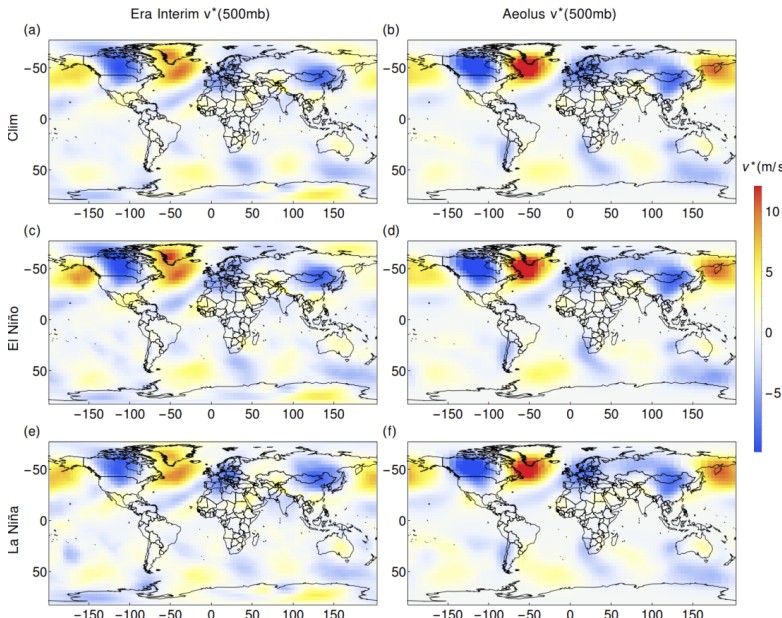

**Figure 3 Monthly mean azonal large scale meridional wind velocity v\* in February at 500mb (compare Fig. 1).**

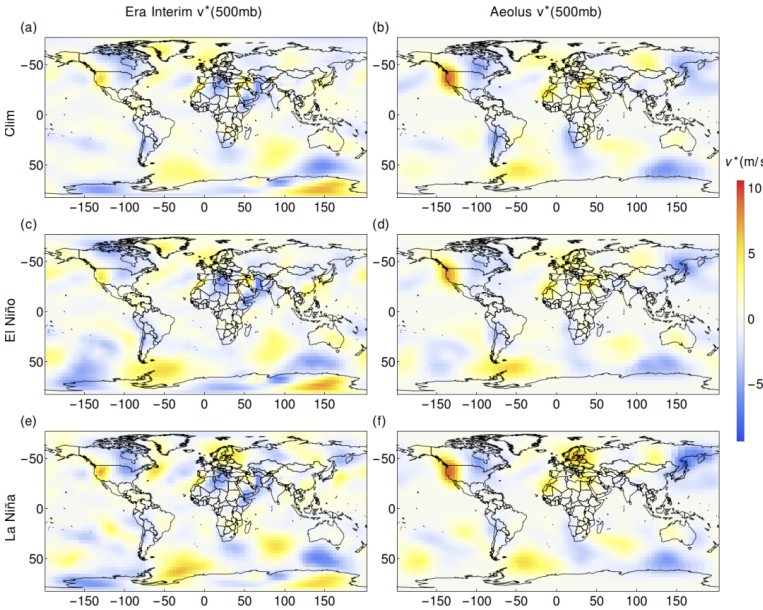

**Figure 4 Monthly mean azonal large scale meridional wind velocity v\* in August at 500mb (compare Fig. 1).**





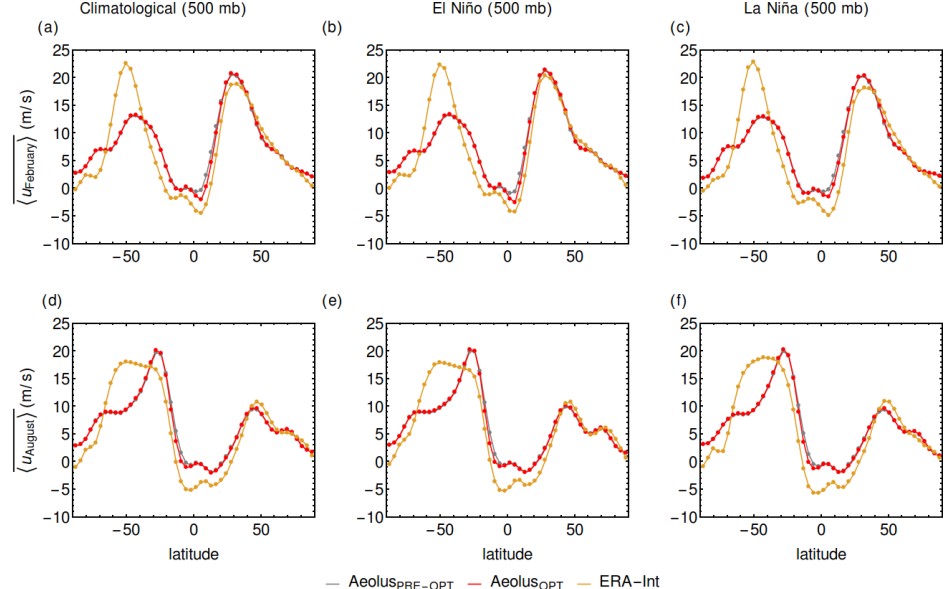

**Figure 5 Monthly mean zonal-mean large scale zonal wind velocity $\overline{\langle u(z,\phi)\rangle}$ at 500mb. (a) shows the monthly mean climatological zonal-mean zonal velocity in February, (b) depicts the monthly mean of el niño months zonal-mean velocity in February and (c) the monthly mean of el niña months zonal-mean velocity in February. (d) displays the monthly mean climatological zonal-mean zonal velocity in August, (e) the monthly mean of el niño months zonal-mean velocity in August and (f) the monthly mean of el niña months zonal-mean velocity in August.**



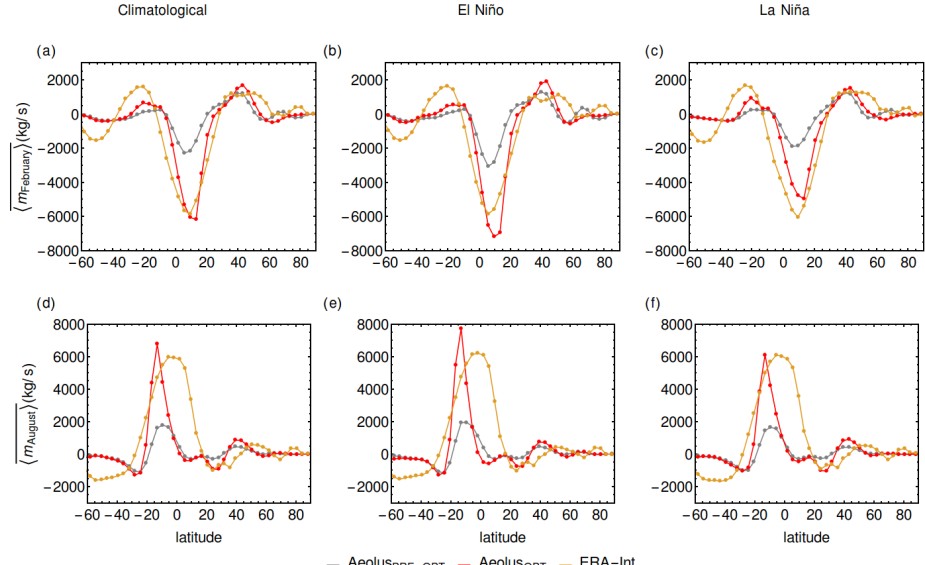

**Figure 6 Monthly mean zonal-mean large scale mass flux $\overline{\langle m \rangle}$. (a) shows the monthly mean climatological zonal-mean mass flux in February, (b) depicts the monthly mean of el niño months zonal-mean mass flux in February and(c) the monthly mean of el niña months zonal-mean mass flux in February. (d) displays the monthly mean climatological zonal-mean mass flux in August, (e) the monthly mean of el niño months zonal-mean mass flux in August and (f) the monthly mean of el niña months zonal-mean mass flux in August.**





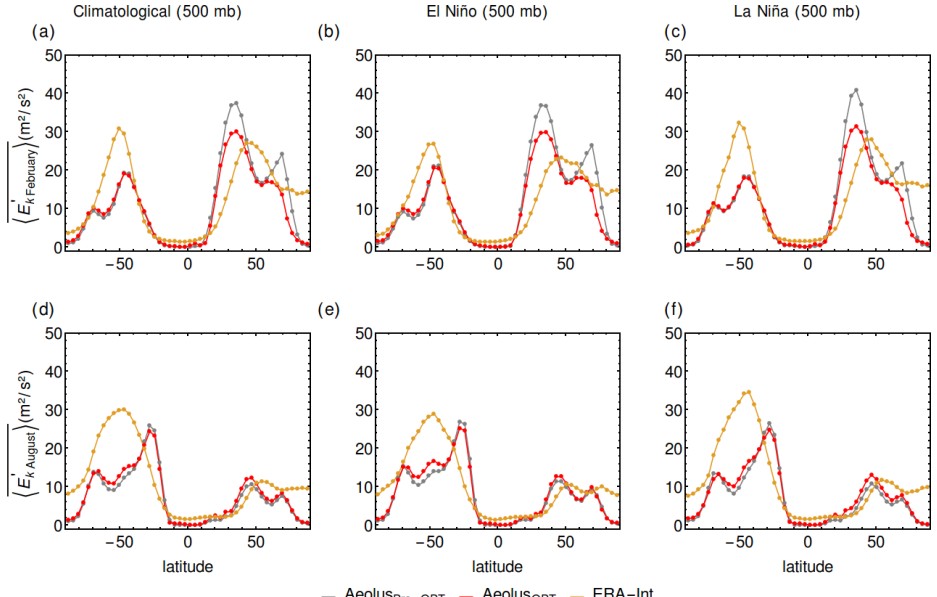

**Figure 7** **Zonal-mean time averaged eddy kinetic energy** $\overline{\langle E'_k \rangle}$ **(compare Fig. 5).**



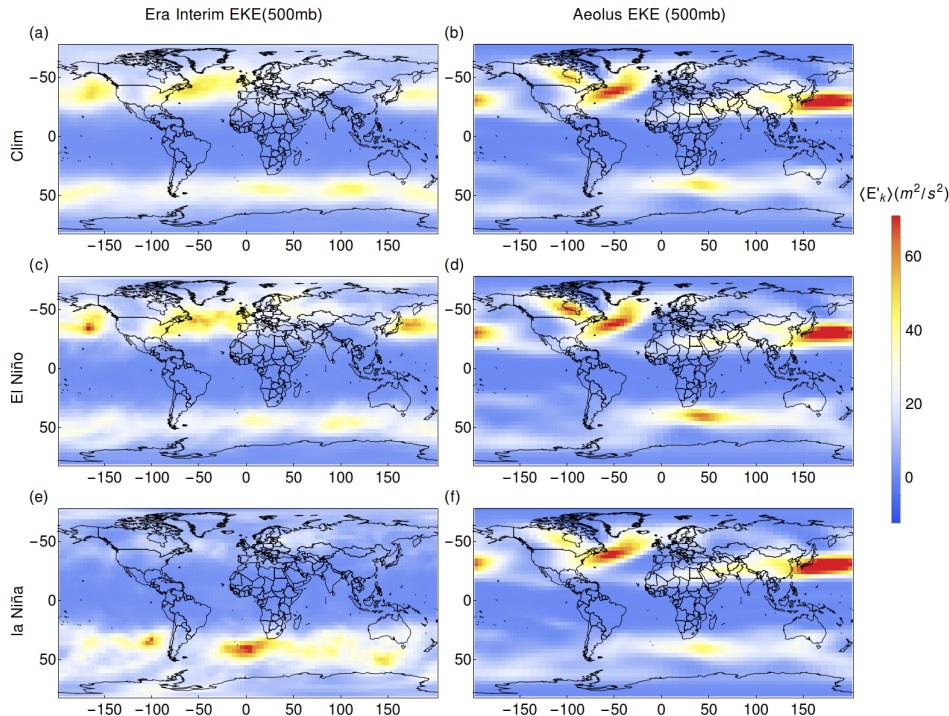

**Figure 8 Monthly mean time averaged eddy kinetic energy $\langle E'_k \rangle$ in February at 500mb (compare Fig. 1).**



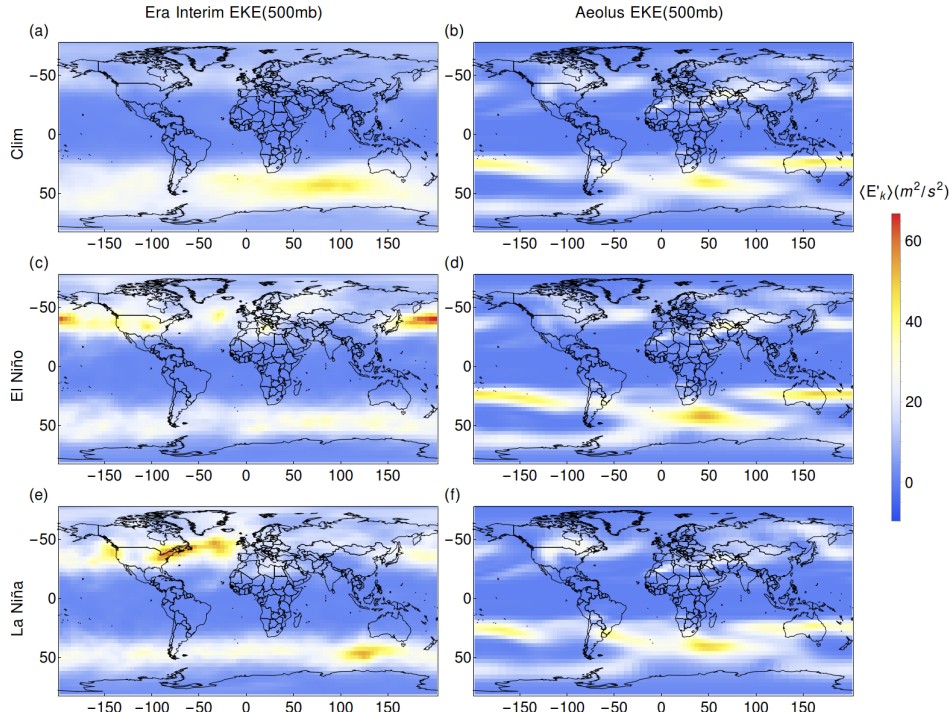

**Figure 9 Monthly mean time averaged eddy kinetic energy $\langle E'_k \rangle$ in August at 500mb (compare Fig. 1).**



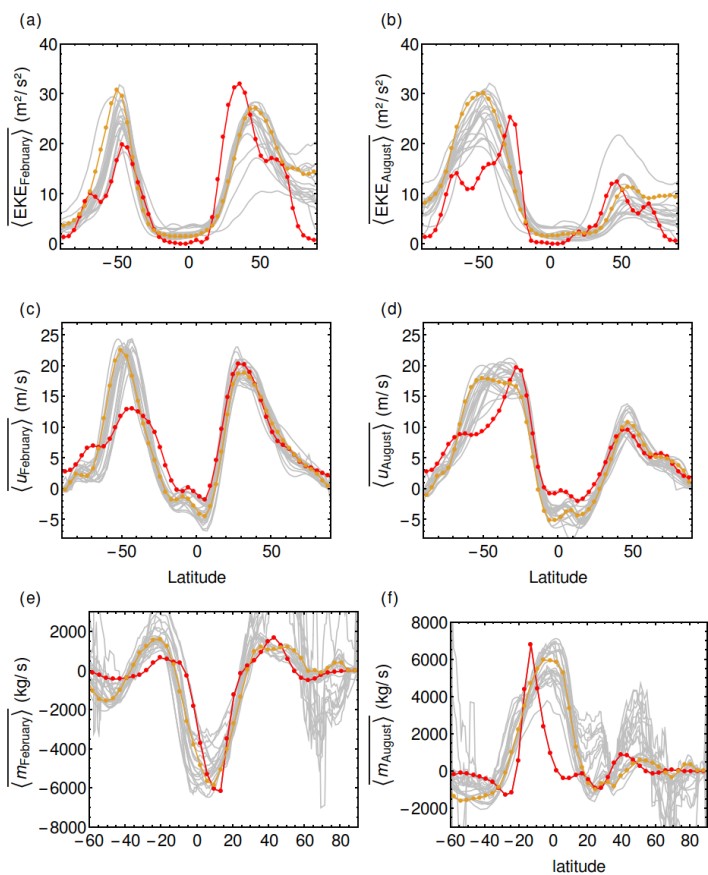

**Figure 10 Comparison to CMIP5-Models. The orange line represents Era Interim data, the red line results from Aeolus and grey lines CMIP5 Models (yearly mean zonal-mean data).**

## 13  Tables

**Table 1 Atmosphere model parameters**

| Symbol | Description | Value |
|---|---|---|
| $a$ | Earth's radius | $6.4 \cdot 10^6$ m |
| $\rho_0$ | Reference air density | 1.3 kg m$^{-3}$ |
| $g$ | gravitational acceleration | 9.8 ms$^{-2}$ |
| $T_0$ | Reference Temperature | 273.16 K |



| $f$ | Coriolis parameter | $2\Omega\sin(\phi)$ |
|---|---|---|
| $\Omega$ | Earth's angular velocity | $7.3 \cdot 10^{-5}\,\mathrm{rad\ s^{-1}}$ |
| $C_\alpha$ | Ageostrophic velocity parameter | 5 |
| $\alpha$ | cross-isobar angle | $\leq 10°$ |
| $H_0$ | Atmosphere scale height | $8 \cdot 10^3\,\mathrm{m}$ |
| $L$ | Latent heat of evaporation | $2.257 \cdot 10^6\,\frac{\mathrm{J}}{\mathrm{Kg}}$ |
| $\Gamma_a$ | Dry adiabatic Lapse Rate | $9.8 \cdot 10^{-3}\,\frac{\mathrm{K}}{\mathrm{m}}$ |
| $\Gamma_0$ | Temperature lapse rate parameters | $5.2 \cdot 10^{-3}\,\frac{\mathrm{K}}{\mathrm{m}}$ |
| $\Gamma_1$ | Temperature lapse rate parameter | $5.5 \cdot 10^{-5}\,\frac{1}{\mathrm{m}}$ |
| $\Gamma_2$ | Temperature lapse rate parameters | $10^{-3}\,\frac{\mathrm{K}}{\mathrm{m}}$ |
| $a_q$ | Temperature lapse rate parameters | $10^3\left(\frac{\mathrm{kg}}{\mathrm{kg}}\right)^2$ |
| $K_z$ | coefficient of the small-scale and meso-scale turbulent exchange for the momentums | $0.005\,z\,\frac{m^2}{s}$ |

**Table 2 Pre-optimized and optimized Parameter set and parameter ranges for optimization step 1**

| Parameters | Optimized value | Range | Pre-optimized value |
|---|---|---|---|
| $\phi_{flat0}$ | 56.5 | 56.0:84.0 | 70.0 |
| $\beta_{NH}$ | 57.2 | 30.0:60.0 | 37.5 |
| $\beta_{SH}$ | -31.3 | -60.0:-30.0 | -52.5 |
| $\Psi_0$ | 10.14 | 7.4 : 12 | 8.0 |



**Table 3 Pre-optimized and optimized Parameter set and parameter ranges for optimization step 2**

| Parameters | optimized value | Range | Pre-optimized value |
|:---:|:---:|:---:|:---:|
| U0 | 5.86 | 3.5:6.5 | 5 |
| m | 0.7849 | 0.4662:0.86658 | 0.6666 |
| d1 | 0.41 | 0.:2. | 1.0 |
| d2 | 2.36 | 0.:2.5 | 1.0 |
| d3 | 0.83 | 0.:2.5 | 1.0 |
| d4 | 1.84 | 0.:2.5 | 1.0 |
| n1 | 2.16 | 0.:2.5 | 1.0 |
| n2 | 1.63 | 0.:2. | 1.0 |
| n3 | 1.06 | 0.:2. | 0.5 |