# Peer review of "The dynamical core of the Aeolus 1.0 Statistical-Dynamical Atmosphere Model: Validation and Parameter Optimization"

_Geoscientific Model Development, 2016_

## Short Comment (SC1) · 5 Dec 2016

Dear authors,

in my role as Executive editor of GMD, I would like to bring to your attention our Editorial version 1.1:

http://www.geosci-model-dev.net/8/3487/2015/gmd-8-3487-2015.html

This highlights some requirements of papers published in GMD, which is also available on the GMD website in the 'Manuscript Types' section:

http://www.geoscientific-model-development.net/submission/manuscript_types.html

[Figure]

In particular, please note that for your paper, the following requirement has not been met in the Discussions paper:

- "The main paper must give the model name and version number (or other unique identifier) in the title."

Please add a version number for Aeolus in the title upon your revised submission to GMD.

Yours,

Astrid Kerkweg

---

## Referee Comment (RC1) · Anonymous Referee #1 · 23 Jan 2017

I have read this article with interest and not without effort. I think it is a valuable endeavor to try and reduce the complexity of models, both for the stated interest of studying very long time variability, and - I would add - to make the physical processes more transparent and easy to interpret.

This having said, I think the manuscript is very elliptic and suffers a real problem of clarity and presentation. I'm ill at ease because at many times I had real problems of understanding. It can be me, of course, but maybe other readers will be in the same situation.

Equations 3, 4 and 8-10, give a diagnostic value of the 5 variables, given a forcing field of temperature and humidity. These are not prognostic equations, they don't give a

time evolution, despite what said in section 2.1. This is ok, but then why are figures 1 to 5 "monthly means" : they would each show field diagnosed from the forcing fields specified in section3. There is a time evolution for the transient kinetic energy u and v and of the momentum flux, <v'u'>, indeed, so I don't understand how these articulate with the diagnostic equations.

Is the above correct? In any case that's what one understands. If so, it should be stated explicitly. The captions of figures 1 to are a bit in contradiction to this though. If I misunderstood, then the things should be explained better. In fact the 2D equations of Pethoukov et al (2000) for T and humidity are progonstic equations, but they are just mentioned at the beginning. Are you integrating these equations along with the equations of the kinetic energies? This is not what it seems to be implied at page 6 line 5. And also, if so, how does forcing comes in?

As you see these are all very basic doubts that clearly come form a bad structuring of the paper. Note also that the supplementary material is not well articulated with the text. The text should contain enough information to understand the basics (like my doubts above). As for now, the derivation of the equations are divided in the two parts - test and supplement - in a chaotic way. Also note that a section 2 of sup. material is referenced in the text, but it's not in there.

In addition to the clarity problem, which is in itself bad enough to require a major revision of the article, there is another point that is not clear to me. The aeolus model as it is presented has already been published in Coumou et al 2011. Is the coupling with the convection model , or the coupling with the temperature and humidity 2D equations of Pethoukov et al (2000) the novelty? Is it the optimization of parameters? Please state this clearly. I have to say that the optimization does not appear to have such a major impact to me. Note also that the method of optimization (simulated annealing) should at least be schematically described.

Below - as a help - are a few specific indications on clear problems of the text, they are

not comprehensive at all. The above considerations should also be addressed. After that is done, a more in depth assessment of the scientific interest of the paper will be more doable.

page 2 line 32 "convective plus 3 layer stratiform" What does this mean?

Section S1.2 "With $K_z = 005$ and In (4 )" incomprehensible

page 3 of s Supp. mat. at the bottom. Is the independency of the large scale and synoptic waves a reasonable assumption? Comment.

Repetition page 4 sup mat. Paragraphe "The contribution to the vertical….",

Page 4 of Supp. Mat. The scale analysis attests… have you done the scale analysis, or is taken from literature?

Page 3, eq.3, could we call it geostrophic and thermal and balance?

Page 3, formula for the meridional pressure, where does that come from? Please describe it more carefully.

page 3 line 9 "Supl.Ment"

Page 3 line 25 repetition, reword.

page 4 line 5. In fact the parameters gamma and $a_q$ are not at all explained in the table. just listed along with their values.

pag 4 line7 is $n_c$ constant or is it computed? If it is a constant, what's it value?

pag 4 line 9, is $U_{sf}$ the same as $U\_Sprofile$ in the supplementary material? if so, it is not clearly explained, what does "The additional calculating of $U\_sprofile$ instead of using the calculated surface zonal velocity is done to avoid instabilities." mean?

Page 4 last line. There is no S.2 in the supplementary material.

page 5 line 19 "equipartitioned

in the supplementary material, the explanation of eq.4 is not complete, it is not shown why the introduction f coefficients d1 d2 and d3 is necessary and how they are chosen.

Note also that the supplementary material is not references, page numbers, line numbers. . ..

---

## Author Comment (AC1) · 17 May 2017

We thank the reviewer for the time she/he took and for the very helpful comments provided, which will help us to improve the manuscript. A pointwise reply to the reviewer's comment is given below.

1.) *Equations 3, 4 and 8-10, give a diagnostic value of the 5 variables, given a forcing field of temperature and humidity. These are not prognostic equations, they don't give a time evolution, despite what said in section 2.1.  This is ok, but then why are figures 1 to 5 "monthly means": they would each show field diagnosed from the forcing fields specified in section 3. There is a time evolution for the transient kinetic energy u and v and of the momentum flux, <v'u'>, indeed, so I don't understand how these articulate with the diagnostic equations. Is the above correct?*

Yes, Eqs. 3, 4, 8 - 10 are not prognostic equations. They describe how the state of the model is calculated from the input data (surface temperature, humidity, and cumulus cloud amount). The input data is given as monthly mean data, therefore figures 1 to 5 show the mean state of the model for a given month.
The transient kinetic equations are also diagnostic equations and the completely derivation of the diagnostic equations has been described in Coumou et al. (2011 ).

We have rewritten the manuscript to state this more clearly.

2.) *In fact the 2D equations of Petoukhov et al (2000) for temperature and humidity are prognostic equations, but they are just mentioned at the beginning. Are you integrating these equations along with the equations of the kinetic energies? This is not what it seems to be implied at page 6 line 5. And also, if so, how does forcing comes in?*

No, we are not integrating the equations for temperature and humidity.
The scope of this study is to describe and test a new set of equations of the dynamical core only. The prognostic equations for T and humidity and the diagnostic equations for EKE have been described and validated in previous work (notably Petoukhov et al, 2000 and Coumou et al, 2011).

Therefore, in order to make the manuscript not larger than needed, we prefer to only reference those publications, rather than providing the full derivation again.

We will rewrite this part to make it clearer.

3.) *As you see these are all very basic doubts that clearly come from a bad structuring of the paper. Note also that the supplementary material is not well articulated with the text. The text should contain enough information to understand the basics (like my doubts above). As for now, the derivation of the equations are divided in the two parts -test and supplement - in a chaotic way. Also note that a section 2 of sup. material is referenced in the text, but it's not in there*

We agree with the reviewer that we did not sufficiently explain our approach. Therefore we have rewritten the main text such that it has enough information to understand our general approach and also better link it to information in the Suppl. Mat. .
We have described the model setup and the experiment in more detail and also have explained already in the abstract the novelty of our model.
In addition, we added Suppl. Mat. 2.

4.) *In addition to the clarity problem, which is in itself bad enough to require a major revision of the article, there is another point that is not clear to me. The Aeolus model as it is presented has already been published in Coumou et al 2011. Is the coupling with the convection model , or the coupling with the temperature and humidity 2D equations of Petoukhov et al (2000) the novelty? Is it the optimization of parameters? Please state this clearly.  I have to say that the optimization does not appear to have such a major impact to me.*

From a theoretical point of view, the novelty is the newly derived statistical dynamical equations of the large-scale zonal-mean field and the planetary waves, and their embedding in the model's dynamical core.

From a technical point of view, we present a detailed parameter-optimization scheme to validate the dynamical core against observations. This is also presented the first time.

5.) *Note also that the method of optimization (simulated annealing) should at least be schematically described.*

We will provide the following schematic plot of the optimization process in the Suppl. Mat. as well as an additional reference in the main text (Kirkpatrick, 1984).

[Figure]

**Figure 1 Schematic plot of the optimization process: The dynamical core is calculated for given parameters (presented in Table 1). In order to find the optimized parameters, we cailbrate the dynamical core with Simulated Annealing and using ERA-Int data to construct the skill function.**

6.) *page 2 line 32 "convective plus 3 layer stratiform" What does this mean?*

It means, that our cloud model simulates 3 types of stratiform clouds, at low-level, mid-level, and upper-level (as described in detail in Eliseev et al, 2013).
 The fourth cloud type represents convective (cumulus) clouds. In the equations for the dynamical core, only cumulus clouds are considered. We will clarify this in the text.

7.) *Section S1.2 "With K_z = 005 and In (4 )" incomprehensible*

We will change that sentence to: With $K_z = 0.005\ z$ and

$$A = \frac{\mathcal{L}\overline{\langle P_{co} \rangle}}{H_0} \frac{\overline{\langle u_{sf} \rangle}}{\Gamma_a - \Gamma_0 - \Gamma_1(T_a - T_0)\left(1 - a_q q_s^2\right) + \Gamma_2 n_c}$$

8.) *Supp. mat. at the bottom. Is the independency of the large scale and synoptic waves a reasonable assumption? Comment.*

This sentence was phrased incorrectly. Large scale und synoptic waves are not independent. Due to a "gap" in the three-dimensional (period-wavelength-phase velocity) spectrum of atmospheric processes (see, e.g., Fraedrich & Böttger 1978, Coumou et al. 2011), the interaction of the synoptic-scale wind component with the large scale long-term wind component (on time scales of about 10-20 days and longer) could to a first approximation, be represented in terms of its ensemble (statistical) characteristics (the second and higher-order moments), and not in terms of the individual eddies (Saltzman, 1978).
Because of this gap between the two spectra, it can be assumed that the long-term component is nearly constant over synoptic timescales.
Hence, the equation can be written as:
<xy'>=x<y'>=x*0=0, again this is explained in detail in Coumou et al. 2011.

We will rewrite this part to make it clearer.

9.) *Repetition page 4 sup mat. Paragraph "The contribution to the vertical…"*

We will remove the repetition.

10.) *Page 4 of Supp. Mat. The scale analysis attests,have you done the scale analysis,or is taken from literature?*

This scaling analysis is described in:
Petoukhov V, Ganopolski A, Claussen M (2003) POTSDAM - a set of atmosphere statistical-dynamical models: theoretical background. Potsdam-Institut für Klimafolgenforschung, ISSN 1436-0179, 136 pp, http://www.pikpotsdam.de/research/publications/pikreports/.files/pr81.pdf

Using the magnitude analysis

$$\langle \overline{w} \rangle \frac{\partial \langle u \rangle}{\partial z} = H^* \left(\frac{u^*}{L^*}\right)\left(\frac{\langle u \rangle}{H}\right) \ll 1$$

where
H = 10^4m is the atmospheric density vertical scale,
u* =10 m/s is characteristic scales of the planetary wave velocities
L*= 3*10^6 are the characteristic scales of planetary horizontal lengths, and H*=H Ro*, where Ro*=u*/(L*f) is the Rossby number for the planetary waves.

11.) *Page 3, eq.3, could we call it geostrophic and thermal wind balance?*

Yes, it could be called thermal wind balance, we will add a sentence in the manuscript.

12.) *Page 3, formula for the meridional pressure, where does that come from? Please describe it more carefully.*

It is derived from Pethoukov et al. (2000) eq. (13). In order to make the manuscript not larger than needed, we would like to only reference those equations, derived in other publications rather describe them again.

13.) page 3 line 9 "Supl.Ment"

We will rewrite as suggested.

14.) Page 3 line 25 repetition, reword.

We will remove the repetition.

15.) *page 4 line 5. In fact the parameters gamma and a_q are not at all explained in the table. just listed along with their value*

Gamma is the lapse rate equation, which is assumed to be linear (Petoukhov et al, 2000):
$$\Gamma = \Gamma_0 + \Gamma_1 (T_a - T_0)(1 - a_q q_s^2) - \Gamma_2 n_c$$
We will add this in the manuscript.
The parameters are all described in Table 1.

16.) *pag 4 line7 is n_c constant or is it computed? If it is a constant, what's it value?*

We used observational data: Multi-year averages of monthly mean, El Niño and La Niña months cumulative cloud fraction is taken from ISSCP (Rossow and Commission, 1996). We will add a sentence in the manuscript to make that clearer.

17.) *pag 4 line 9, is U_sf the same as U_Sprofile in the supplementary material? if so, it is not clearly explained, what does "The additional calculating of U_sprofile instead of using the calculated surface zonal velocity is done to avoid instabilities." mean?*

Yes, U_sf is the same as Us_profile. We will rewrite that in the manuscript. Instabilities can emerge due to the strong positive feedback between the meridional temperature profile, surface winds and vertical wind velocity, which can lead to high latent heat release (moisture-convection feedback) (Grabowski and Moncrieff, 2004). In nature these would be damped out but due to the fixed troposphere height in the model, we have to parameterize it.

18.) *Page 4 last line. There is no S.2 in the supplementary material*

We will include Suppl. Mat. 2.

19.) page 5 line 19 "equipartitioned

We will rewrite it.

20.) *in the supplementary material, the explanation of eq.4 is not complete, it is not shown why the introduction of coefficients d1 d2 and d3 is necessary and how they are chosen*

The derived equation for the meridional velocity does not account for latent heat release associated with convective precipitation. To capture this additional term we include convective precipitation and finally introduce tuning parameters, which have values close to 1.
We will add that part to explain it better.

21.) *Page 4 last line. There is no S.2 in the supplementary material*

We will include it.

22.) *Note also that the supplementary material is not references, page numbers, line numbers*

We will change that in the updated manuscript.

---

## Referee Comment (RC2) · Anonymous Referee #2 · 17 Oct 2017

While reviewing this article I have looked through both the comments of the other reviewer and the authors' response. I agree with almost all of the other reviewer's comments and I am pleased to see that the authors have responded with changes to their manuscript. For this reason I will try to not repeat comments made in the other review that have already been addressed.

I think that the material within this article is good, however I found it difficult to read and some parts hard to understand. As a general point, I think the authors should take time to try and make the article more accessible. Below are a few more specific comments:

1) I think that the introduction needs more information about statistical dynamical atmosphere models. For readers from a dynamical core background it would be very useful to give a brief paragraph about how they work and the key differences to dynamical cores.

2) The introduction should also better explain the aims of the paper. This is covered briefly in the abstract, but it would be worth including this in the introduction.

3) The way the article is worded it is not clear if the work presented is a completely new model, or something coupled to an existing model (which is what I think is the case). This needs to be clarified.

4) Page 8, line 7, you exclude the polar regions. I am curious why it is poleward of 60 degrees for SH but 70 degrees for NH. Why the difference?

5) In the abstract, page 1 line 18, change "enables us to do climate simulations" to "enables us to perform climate simulations"

6) Typo, page 7 line 29. Full stop needed after the reference.

---

## Author Response (AR1)

We thank the reviewers for the time they took and for the very helpful comments provided, which will help us to improve the manuscript. A pointwise reply to the reviewer's comment is given below.

**Reviewer #1:**

1.) *Equations 3, 4 and 8-10, give a diagnostic value of the 5 variables, given a forcing field of temperature and humidity. These are not prognostic equations, they don't give a time evolution, despite what said in section 2.1. This is ok, but then why are figures 1 to 5 "monthly means": they would each show field diagnosed from the forcing fields specified in section 3. There is a time evolution for the transient kinetic energy u and v and of the momentum flux, <v'u'>, indeed, so I don't understand how these articulate with the diagnostic equations. Is the above correct*?

Yes, Eqs. 3, 4, 8 - 10 are not prognostic equations. They describe how the state of the model is calculated from the input data (surface temperature, humidity, and cumulus cloud amount). The input data is given as monthly mean data, therefore figures 1 to 5 show the mean state of the model for a given month.
The transient kinetic equations are also diagnostic equations and the completely derivation of the diagnostic equations has been described in Coumou et al. ( 2011).

We have rewritten the manuscript to state this more clearly (p .4, l. 12-21, p. 7,l. 1-4).

2.) *In fact the 2D equations of Petoukhov et al (2000) for temperature and humidity are prognostic equations, but they are just mentioned at the beginning. Are you integrating these equations along with the equations of the kinetic energies? This is not what it seems to be implied at page 6 line 5. And also, if so, how does forcing comes in?*

No, we are not integrating the equations for temperature and humidity.
The scope of this study is to describe and test a new set of equations of the dynamical core only. The prognostic equations for T and humidity and the diagnostic equations for EKE have been described and validated in previous work (notably Petoukhov et al, 2000 and Coumou et al, 2011).

Therefore, in order to make the manuscript not larger than needed, we prefer to only reference those publications, rather than providing the full derivation again.

We rewrote this part to make it clearer (p. 3, l. 24-30; p. 4, l. 12-21; p. 7,l. 1-4; p. 7, l. 14 – p. 8, l. 4).

3.) *As you see these are all very basic doubts that clearly come from a bad structuring of the paper. Note also that the supplementary material is not well articulated with the text. The text should contain enough information to understand the basics (like my doubts above). As for now, the derivation of the equations are divided in the two parts -test and supplement - in a chaotic way. Also note that a section 2 of sup. material is referenced in the text, but it's not in there*

We agree with the reviewer that we did not sufficiently explain our approach. Therefore we have rewritten the main text such that it has enough information to understand our general approach and also better link it to information in the Suppl. Information.

We have described the model setup and the experiment in more detail and also have explained already in the abstract the novelty of our model (p. 1, l.21 – p.2, l. 2; p.3, l. 24 – 36; p. 4, l.12 - 21).
In addition, we added Suppl. Mat. 2 (in Suppl. Inf.: now S1.3 Zeroth order solution of the thermally induced waves of the barotropic vorticity equation at the EBL; p.2, l. 7 - p.3,l. 8).

4.) *In addition to the clarity problem, which is in itself bad enough to require a major revision of the article, there is another point that is not clear to me. The Aeolus model as it is presented has already been published in Coumou et al 2011. Is the coupling with the convection model , or the coupling with the temperature and humidity 2D equations of Petoukhov et al (2000) the novelty? Is it the optimization of parameters? Please state this clearly.  I have to say that the optimization does not appear to have such a major impact to me.*

From a theoretical point of view, the novelty is the newly derived statistical dynamical equations of the large-scale zonal-mean field and the planetary waves, and their embedding in the model's dynamical core.

From a technical point of view, we present a detailed parameter-optimization scheme to validate the dynamical core against observations. This is also presented the first time.

5.) *Note also that the method of optimization (simulated annealing) should at least be schematically described.*

We provided the following schematic plot of the optimization process in the Suppl. Mat. as well as an additional reference in the main text (Kirkpatrick, 1984) (in Suppl. Inf.: p. 6, l. 10).

[Figure]

**Figure 1 Schematic plot of the optimization process: The dynamical core is calculated for given parameters (presented in Table 1). In order to find the optimized parameters, we calibrate the dynamical core with Simulated Annealing and using ERA-Int data to construct the skill function.**

6.) *page 2 line 32 "convective plus 3 layer stratiform" What does this mean?*

It means, that our cloud model simulates 3 types of stratiform clouds, at low-level, mid-level, and upper-level (as described in detail in Eliseev et al, 2013).

The fourth cloud type represents convective (cumulus) clouds. In the equations for the dynamical core, only cumulus clouds are considered. We clarified this in the text. (p.3, l. 38-p. 4, l. 2)

7.) *Section S1.2 "With K_z = 005 and In (4 )" incomprehensible*

We changed that sentence to: With $K_z = 0.005\ z$ and

$$A = \frac{\mathcal{L}\overline{\langle P_{co}\rangle}}{H_0} \frac{\overline{\langle u_{sf}\rangle}}{\Gamma_a - \Gamma_0 - \Gamma_1(T_a - T_0)\left(1 - a_q q_s^2\right) + \Gamma_2 n_c}$$

(in Suppl. Information, p. 3, l. 11 - 13)

8.) *Supp. mat. at the bottom. Is the independency of the large scale and synoptic waves a reasonable assumption? Comment.*

This sentence was phrased incorrectly. Large scale und synoptic waves are not independent. Due to a "gap" in the three-dimensional (period-wavelength-phase velocity) spectrum of atmospheric processes (see, e.g., Fraedrich & Böttger 1978, Coumou et al. 2011), the interaction of the synoptic-scale wind component with the large scale long-term wind component (on time scales of about 10-20 days and longer) could to a first approximation, be represented in terms of its ensemble (statistical) characteristics (the second and higher-order moments), and not in terms of the individual eddies (Saltzman, 1978).
Because of this gap between the two spectra, it can be assumed that the long-term component is nearly constant over synoptic timescales.
Hence, the equation can be written as:
<xy'>=x<y'>=x*0=0, again this is explained in detail in Coumou et al. 2011.

We rewrote this part to make it clearer (in Suppl. Information, p. 5, l. 5 - 13).

9.) *Repetition page 4 sup mat. Paragraph "The contribution to the vertical…"*

We removed the repetition.

10.) *Page 4 of Supp. Mat. The scale analysis attests,have you done the scale analysis,or is taken from literature?*

This scaling analysis is described in:
Petoukhov V, Ganopolski A, Claussen M (2003) POTSDAM - a set of atmosphere statistical-dynamical models: theoretical background. Potsdam-Institut für Klimafolgenforschung, ISSN 1436-0179, 136 pp, http://www.pikpotsdam.de/research/publications/pikreports/.files/pr81.pdf

Using the magnitude analysis

$$\langle \overline{w}\rangle \frac{\partial \langle u\rangle}{\partial z} = H^* \left(\frac{u^*}{L^*}\right)\left(\frac{\langle u\rangle}{H}\right) \ll 1$$

where
H = 10^4m is the atmospheric density vertical scale,
u* =10 m/s is characteristic scales of the planetary wave velocities

L\*= 3\*10^6 are the characteristic scales of planetary horizontal lengths, and H\*=H Ro\*, where Ro\*=u\*/(L\*f) is the Rossby number for the planetary waves.

*11.)Page 3, eq.3, could we call it geostrophic and thermal wind balance?*

Yes, it could be called thermal wind balance, we added a sentence in the manuscript (p. 4, l. 31).

*12.)Page 3, formula for the meridional pressure, where does that come from? Please describe it more carefully.*

It is derived from Petoukhov et al. (2000) eq. (13). In order to make the manuscript not larger than needed, we would like to only reference those equations, derived in other publications rather describe them again (p. 5, l. 3).

*13.)page 3 line 9 "Supl.Ment"*

We rewrote as suggested.

*14.)Page 3 line 25 repetition, reword.*

We removed the repetition.

*15.)page 4 line 5. In fact the parameters gamma and a_q are not at all explained in the table. just listed along with their value.*

Gamma is the lapse rate equation, which is assumed to be linear (Petoukhov et al, 2000):
$$\Gamma = \Gamma_0 + \Gamma_1(T_a - T_0)\left(1 - a_q q_s^2\right) - \Gamma_2 n_c$$
We added this in the manuscript (p. 6, l. 14).
The parameters are all described in Table 1.

*16.)pag 4 line7 is n_c constant or is it computed? If it is a constant, what's it value?*

We used observational data: Multi-year averages of monthly mean, El Niño and La Niña months cumulative cloud fraction is taken from ISSCP (Rossow and Schiffer, 1999). We added a sentence in the manuscript to make that clearer (p. 5, l. 15-16; p. 8, l. 13 - 15).

*17.)pag 4 line 9, is U_sf the same as U_Sprofile in the supplementary material?  if so, it is not clearly explained, what does "The additional calculating of U_sprofile instead of using the calculated surface zonal velocity is done to avoid instabilities." mean?*

Yes, U_sf is the same as Us_profile. We will rewrite that in the manuscript (p. 5, l. 17; in Suppl. Inf.: p. 4, l. 1 - 5).
Instabilities can emerge due to the strong positive feedback between the meridional temperature profile, surface winds and vertical wind velocity, which can lead to high latent heat release (moisture-convection feedback) (Grabowski and Moncrieff, 2004). In nature these would be damped out but due to the fixed troposphere height in the model, we have to parameterize it.

*18.)Page 4 last line. There is no S.2 in the supplementary material*

We included Suppl. Inf. 2 (in Suppl. Inf.: S1.3 p. 2, l. 7 - p. 3,l. 6)..

*19.)page 5 line 19 "equipartitioned*

We rewrote it (p. 7, l. 6).

*20.)in the supplementary material, the explanation of eq.4 is not complete, it is not shown why the introduction of coefficients d1 d2 and d3 is necessary and how they are chosen*

The derived equation for the meridional velocity does not account for latent heat release associated with convective precipitation. To capture this additional term we include convective precipitation and finally introduce tuning parameters, which have values close to 1.
We added that part to explain it better (in Suppl. Inf. p. 6, l. 5 - 7).

*21.)Page 4 last line. There is no S.2 in the supplementary material*

We included Suppl. Inf. 2 (in Suppl. Inf.: S1.3 p. 2, l. 7 - p. 3,l. 8)..

*22.)Note also that the supplementary material is not references, page numbers, line numbers*

We changed that in the updated manuscript.

**Reviewer #2:**

*1.) While reviewing this article I have looked through both the comments of the other reviewer and the authors' response. I agree with almost all of the other reviewer's comments and I am pleased to see that the authors have responded with changes to their manuscript. For this reason I will try to not repeat comments made in the other review that have already been addressed. I think that the material within this article is good, however I found it difficult to read and some parts hard to understand. As a general point, I think the authors should take time to try and make the article more accessible. Below are a few more specific comments.*

Thanks, we rewrote those parts in order to make the article more accessible as outlined below.

*2.) I think that the introduction needs more information about statistical dynamical atmosphere models. For readers from a dynamical core background it would be very useful to give a brief paragraph about how they work and the key differences to dynamical cores.*

We agree with the reviewer and included the following information about SDAMs (p. 3, l. 4 - 23):

In particular, statistical-dynamical (SD) atmosphere models parameterize smaller scale (and more short-lived) processes like synoptic eddy activity in terms of the large-scale, long-term fields. The assumption of those models is thus that atmospheric variables can be expressed in separate terms of a large-scale, long-term component, with characteristic spatial and temporal scales of L>1000 km and T>10 days, and a small-scale component like ensembles of synoptic-scale eddies and waves. The latter are then parameterized by their averaged statistical

characteristics (e.g. their total kinetic energy and heat, moist and momentum fluxes). This means that transport effects of the fast moving weather systems on the large-scale, long-term atmospheric motion are averaged (Ehlers et al., 2001).

The essential difference to GCMs is thus the point of truncation in the frequency spectrum of atmospheric motion (Saltzman, 1978). GCMs solve all phenomena of frequencies lower than and including synoptic cyclones (and sometimes even mesoscale systems), whereas SD models parameterize all scales smaller and equal to synoptic. Much of the difficulty in SD models is to define physically reasonable parameterizations occurring in the equations (Saltzman, 1978). For Aeolus the synoptic parameterization has been described in detail in Coumou et al. (2011).

As written above, SD models are also spatially averaged since for long-term climate simulations, we are typically interested in the large spatial aspects of the climate. It is further practical to split the large-scale, long-term field into two components: the zonally averaged mean field, and the asymmetric departure of the field from the zonally averaged fields characterizing the East-West variations. The azonal variables can be, for example, resolved by 1-dimensional Fourier components around latitude circles or into spherical harmonics (Saltzman, 1978).

3.) *The introduction should also better explain the aims of the paper. This is covered briefly in the abstract, but it would be worth including this in the introduction. The way the article is worded it is not clear if the work presented is a completely new model, or something coupled to an existing model (which is what I think is the case). This needs to be clarified.*

We agree with the reviewer and will include the aims of this paper in the introduction as well. In addition, we explain the model and the new aspects in more detail. In particular, we wrote (p. 3, l. 24 - 36):

Here, we present the Aeolus 1.0 dynamical core, developed at the Potsdam Institute for Climate Impact Research (PIK), a new SD model for the atmosphere. It uses some aspects of the atmosphere module of the EMIC CLIMBER-2 developed by Petoukhov et al. (2000). The dynamical core is completely new with novel equations for the large-scale meridional wind speed as well as quasi-stationary planetary waves. Together with the synoptic parameterizations presented in Coumou (2011), these equations form the new dynamical core of Aeolus 1.0. The model is coupled with the cloud module consisting of a three-layer stratiform cloud plus convective cloud scheme as presented and validated in Eliseev et al. (2013).

Further, we present the equations of the model and validate the dynamical core using a parameter optimization experiment. Aeolus 1.0 is forced with prescribed surface temperature, surface humidity and cumulus cloud fraction to test the model's performance. In particular we examine the reproduction of the seasonal cycle, and the influence of ENSO of our model and compare relevant dynamical fields of the model output against seasonal means of ERA-Interim reanalysis data (climatology 1983-2009). The effects of parameter tuning are evaluated to improve the performance of the model.

4.) *Page 8, line 7, you exclude the polar regions. I am curious why it is poleward of 60 degrees for SH but 70 degrees for NH. Why the difference?*

We wanted to exclude influences of the Antarctica, which is located at south of 60 degrees. The reanalysis data is spikey over Antarctica. We wrote this in the revised manuscript (p. 9, l. 25-26).

5.) *In the abstract, page 1 line 18, change "enables us to do climate simulations" to "enables us to perform climate simulations*

Thanks, we corrected it (p. 1, l. 18).

*6.) Typo, page 7 line 29. Full stop needed after the reference.*

Thanks, we corrected it (p. 9, l. 12).

**For the waves excited by the orography, the stream function is calculated by**

$$\beta \nabla_\lambda \langle \psi_{or,0,EBL}^* \rangle = -\frac{f}{H_0} \langle w_{or} \rangle + \frac{f^2}{g} \frac{\partial \langle u'v' \rangle^*}{\partial z} \tag{S5}$$

**where $f$ is the Coriolis parameter and $\beta = \nabla_\phi f$ and**

$$w_{or} = \langle u \rangle \nabla_\phi h_{or} + \langle v \rangle \nabla_\lambda h_{or} + a_{std} (\langle u \rangle^2 + \langle v \rangle^2 + \langle u'^2 \rangle + \langle v'^2 \rangle)^{1/2} h_{std}. \tag{S6}$$

**The variable $h_{or}$ describes the grid cell averaged orography height $h_{std}$ the subgrid scale standard deviation of the height of mountains, and $a_{std}$ is an additional tuning parameter.**

The azonal component describes quasi-stationary planetary waves and is subdivided into a geostrophic and ageostrophic term:

$$u^* = u^*_{geos} + u^*_{ageos}$$

$$v^* = v^*_{geos} + v^*_{ageos}$$

"

p. 2, l. 9 added paragraph "**S1.3 Zeroth order solution of the thermally induced waves of the barotropic vorticity equation at the EBL**

We start from the z-projection of the baroclinic vorticity equation, which can be derived from the simplified Navier-Stokes-equation :

$$\bar{u} \frac{\partial}{\partial x} \left( \frac{\partial^2 \langle \Psi^*_{EBL} \rangle}{\partial x^2} + \frac{\partial^2 \langle \Psi^*_{EBL} \rangle}{\partial y^2} \right) + \beta \frac{\partial \langle \psi^* \rangle}{\partial x} = -\frac{\rho}{T_0} \frac{\partial \langle T^*_{EBL} \rangle}{\partial x} \frac{\partial \bar{p}}{\partial y} \frac{1}{\rho_0^2} \qquad (\text{ S7 })$$

with $\beta = \frac{2\Omega}{a} \cos\phi$ , and $\Omega$ is the earth's rotation angular velocity, $a$ is the earth's radius and $\phi$ the latitude.

In Eq. (S7) $\langle \Psi^*_{EBL} \rangle$ is the stream function of the azonal large-scale component at the equivalent barotropic level $z_{EBL}$, $x$ and $y$ are the horizontal and vertical direction, $T_0$ is the constant reference temperature and $\langle T^*_{EBL} \rangle$ is the large-scale long-term azonal temperature at the EBL. The variable $\bar{u}$ is the zonal mean zonal wind velocity, $\rho_0$ stands for the density near surface and $\bar{p}$ is the zonal mean pressure.

For the stream function of the azonal large-scale component of motion at the equivalent barotropic level $z_{EBL}$ we use the ansatz

$$\langle \Psi^*_{EBL} \rangle = \langle \Psi^*_{0,EBL} \rangle + \epsilon \langle \Psi^*_{1,EBL} \rangle + _{...}$$

For the zeroth order approximation, we can neglect higher order derivations of $\Psi$:

$$\beta \frac{\partial \langle \Psi^*_{0,EBL} \rangle}{\partial x} = -\frac{1}{\rho T_0} \frac{\partial \langle T^*_{EBL} \rangle}{\partial x} \frac{\partial}{\partial y} \frac{\int_0^\infty \rho \langle [T(z)] \rangle \, dz}{H_0} \qquad (\text{ S8 })$$

In eq. (S8), we replaced $\bar{p} = \int_0^\infty R\rho \langle [T(z)] \rangle \, dz / H_0$ and $H_0 = RT_0/g$ and $\rho = \rho_0 \exp(-z/H_0)$, $R$ is the gas constant, $\rho$ is the air density, T is the temperature , $H_0$ is the atmospheric scale height, and $g$ the gravity acceleration . Per definition, one can replace the term with

$$\frac{\int_0^\infty p \, dz}{H_0} = \frac{gR}{RT_0} 2 \int_0^{z_{EBL}} \rho \langle [T(z)] \rangle \, dz$$

**Such that**

$$\frac{\partial \langle \psi^*_{0,EBL} \rangle}{\partial x} = -\frac{agR}{2\Omega R T_0^2 \rho_0 \cos\phi} 2 \frac{\partial}{\partial y} \int_0^{z_{EBL}} \rho \langle [T(z)] \rangle \, dz \frac{\partial \langle T^*_{EBL} \rangle}{\partial x}$$

**With latter equation and** $\frac{1}{a}\nabla_\phi = \partial/\partial y$ **, we can then derive**

$$\langle \psi^*_{0,EBL} \rangle = -\langle T^*_{EBL} \rangle \frac{g}{\Omega \rho_0 T_0^2 \cos\phi} \nabla_\phi \int_0^{z_{EBL}} \rho \langle [T(z)] \rangle \, dz$$

"

p. 2, l. 9, changed "**S1.2**" to "**S2**"

p. 2, l. 9 – p. 5, l. 2 reordered equation numbers

p. 2, l. 13 - 17, changed to "**With** $K_z = 0.005\, z$ **and**

$$A = \frac{\mathcal{L}\overline{\langle P_{co} \rangle}}{H_0} \frac{\overline{\langle u_{sf} \rangle}}{\Gamma_a - \Gamma_0 - \Gamma_1(T_a - T_0)(1 - a_q q_s^2) + \Gamma_2 n_c}$$

**whereby the parameters are given in Table 1. We roughly approximate** $\overline{\langle u_{sf} \rangle}$ **by constant profile for this experiment**

$$\overline{\langle u_{sf} \rangle} = \begin{cases} 2, & |\phi| > 40 \\ -2\,\cos\left(\phi\frac{\pi}{40°}\right), & \text{otherwise} \end{cases}. \tag{S10}$$

"

p. 2, l. 19 added paragraph "**The additional calculation of** $\overline{\langle u_{sf} \rangle}$ **instead of the calculated surface zonal velocity is done to avoid instabilities. Instabilities can emerge due to the strong positive feedback between the meridional temperature and vertical wind velocity, which lead to high latent heat. In nature these would be damped out but due to fixed troposphere height, we parameterize it in the above described way.**"

p. 3, l. 2 changed "$V$ "to "$V$"

p. 3, l. 13-14 changed to

"**Due to a "gap" in the three-dimensional (period-wavelength-phase velocity) spectrum of atmospheric processes (see, e.g., Fraedrich & Böttger 1978, Coumou et al. 2011), the synoptic-scale component in its interaction with the large-scale long-term component of the atmospheric fields on the time scales about 10-20 days and longer could be, to a first approximation, represented (described) in terms of its ensemble (statistical) characteristics**

(the second and higher-order moments), and not as the individual eddies (Saltzman, 1978). We can simplify the terms $\langle \overline{u}v' \rangle = \langle u'\overline{v} \rangle = 0$. In addition, it is $\overline{\overline{u}\overline{v}} = \overline{u}\overline{v}$ due to quasi stationarity of both terms. It is also $\langle \frac{dx}{dt} \rangle = 0$ and $\langle \overline{u}\overline{v} \rangle = \langle \overline{u} \rangle \langle \overline{v} \rangle$ since the oscillations of $\overline{u}$ and $\overline{v}$ are very small and independent of each other."

p. 4, l. 6-7 changed to "Also, the scale analysis attests that $\langle \overline{w} \rangle \frac{\partial \langle u \rangle}{\partial z}$ are small (Petoukhov et al., 2003)"

p. 4, l. 16 changed "$\widetilde{\kappa}$ "to "$K_z$"

p. 5, l. 1 changed "$\frac{dK_z}{z}$ "to "$\frac{dK_z}{dz}$"

p. 5, l. 2 added "

**The derived equation for the meridional velocity does not account for latent heat release associated with convective precipitation. To capture this additional term we include convective precipitation and finally introduce tuning parameters, which have values close to 1.**

**S3 Schematic plot of the optimization process**

[Figure]

„

[revised manuscript text omitted]

**S1 Supplementary  **Information**

**S1.1 Planetary Waves**

 **S1.1 Calculation of** planetary waves.  **at tropospheric levels excluding EBL**

At other tropospheric levels than the EBL, the  components  calculated by

$$\langle u_{EBL}^*(z)\rangle = -\nabla_\phi \langle \psi_{EBL}^* \rangle \langle u^*(z)\rangle = -\frac{1}{f\rho_0}\nabla_\phi\langle p_z^*\rangle \tag{S1}$$

$$\langle v_{EBL}^*(z)\rangle = \nabla_\lambda \langle \psi_{EBL}^* \rangle \langle v^*(z)\rangle = \frac{1}{f\rho_0}\nabla_\lambda\langle p_z^*\rangle, \tag{S2}$$

 The azonal component  is computed assuming isothermal expansion of air parcels in planetary waves

$$\langle p_z^*\rangle = \langle p_{EBL}^*\rangle\exp[(z-z_{EBL})/H_0] + \frac{p_* g}{\Gamma R}\exp[-z/H_0]\left\{\ln\left[\frac{T(z)}{T(z_{EBL})}\right] - \ln\left[\overline{\frac{T(z)}{T(z_{EBL})}}\right]\right\} \tag{S3}$$

and

$$\langle p_{EBL}^*\rangle = \rho\left\{\overline{\langle u_{EBL}\rangle}\right\}\nabla_\phi\langle\psi_{EBL}^*\rangle + 2\left(\frac{\overline{\langle u_{EBL}\rangle}}{a\cos(\phi)} + \Omega\right)\sin(\phi)\langle\psi_{EBL}^*\rangle \tag{S4}$$

**S1.1.1  **2 Orographically induced** stream function**

For the waves excited by the orography, the stream function is calculated by

$$\beta\nabla_\lambda\langle\psi_{or,0,EBL}^*\rangle = -\frac{f}{H_0}\langle w_{or}\rangle + \frac{f^2}{g}\frac{\partial\langle u'v'\rangle^*}{\partial z} \tag{S5}$$

where $f$ is the Coriolis parameter and $\beta = \nabla_\phi f$ and

$$w_{or} = \langle u \rangle \nabla_\phi h_{or} + \langle v \rangle \nabla_\lambda h_{or} + a_{std}(\langle u \rangle^2 + \langle v \rangle^2 + \langle u'^2 \rangle + \langle v'^2 \rangle)^{1/2} h_{std}. \tag{S6}$$

The variable $h_{or}$ describes the grid cell averaged orography height $h_{std}$ the subgrid scale standard deviation of the height of mountains, and $a_{std}$ is an additional tuning parameter.

The azonal component describes quasi-stationary planetary waves and is subdivided into a geostrophic and ageostrophic term:

$$u^* = u^*_{geos} + u^*_{ageos}$$

$$v^* = v^*_{geos} + v^*_{ageos}$$

**S1.3 Zeroth order solution of the thermally induced waves of the barotropic vorticity equation at the EBL**

We start from the z-projection of the baroclinic vorticity equation, which can be derived from the simplified Navier-Stokes-equation :

$$\bar{u} \frac{\partial}{\partial x} \left( \frac{\partial^2 \langle \Psi^*_{EBL} \rangle}{\partial x^2} + \frac{\partial^2 \langle \Psi^*_{EBL} \rangle}{\partial y^2} \right) + \beta \frac{\partial \langle \psi^* \rangle}{\partial x} = -\frac{\rho}{T_0} \frac{\partial \langle T^*_{EBL} \rangle}{\partial x} \frac{\partial \bar{p}}{\partial y} \frac{1}{\rho_0^2} \tag{S7}$$

with $\beta = \frac{2\Omega}{a} \cos \phi$, and $\Omega$ is the earth's rotation angular velocity, $a$ is the earth's radius and $\phi$ the latitude.

In Eq. (S7) $\langle \Psi^*_{EBL} \rangle$ is the stream function of the azonal large-scale component at the equivalent barotropic level $z_{EBL}$, $x$ and $y$ are the horizontal and vertical direction, $T_0$ is the constant reference temperature and $\langle T^*_{EBL} \rangle$ is the large-scale long-term azonal temperature at the EBL. The variable $\bar{u}$ is the zonal mean zonal wind velocity, $\rho_0$ stands for the density near surface and $\bar{p}$ is the zonal mean pressure.

For the stream function of the azonal large-scale component of motion at the equivalent barotropic level $z_{EBL}$ we use the ansatz

$$\langle \Psi^*_{EBL} \rangle = \langle \Psi^*_{0,EBL} \rangle + \epsilon \langle \Psi^*_{1,EBL} \rangle + \dots$$

For the zeroth order approximation, we can neglect higher order derivations of $\Psi$:

$$\beta \frac{\partial \langle \Psi_{0,EBL}^* \rangle}{\partial x} = -\frac{1}{\rho\, T_0} \frac{\partial \langle T_{EBL}^* \rangle}{\partial x} \frac{\partial}{\partial y} \frac{\int_0^\infty \rho \langle [T(z)] \rangle\, dz}{H_0} \qquad ( S8 )$$

In eq. (S8), we replaced $\bar{p} = \int_0^\infty R\rho\langle[T(z)]\rangle\, dz/H_0$ and $H_0 = RT_0/g$ and $\rho = \rho_0 \exp(-z/H_0)$, $R$ is the gas constant, $\rho$ is the air density, T is the temperature , $H_0$ is the atmospheric scale height, and $g$ the gravity acceleration . Per definition, one can replace the term with

$$\frac{\int_0^\infty p\, dz}{H_0} = \frac{gR}{RT_0} 2 \int_0^{z_{EBL}} \rho\langle[T(z)]\rangle\, dz$$

Such that

$$\frac{\partial \langle \psi_{0,EBL}^* \rangle}{\partial x} = -\frac{agR}{2\Omega RT_0^2 \rho_0 \cos\phi} 2 \frac{\partial}{\partial y} \int_0^{z_{EBL}} \rho\langle[T(z)]\rangle\, dz \frac{\partial \langle T_{EBL}^* \rangle}{\partial x}$$

With latter equation and $\frac{1}{a}\nabla_\phi = \partial/\partial y$ , we can then derive

$$\langle \psi_{0,EBL}^* \rangle = -\langle T_{EBL}^* \rangle \frac{g}{\Omega \rho_0 T_0^2 \cos\phi} \nabla_\phi \int_0^{z_{EBL}} \rho\langle[T(z)]\rangle\, dz$$

**S2 Derivation of the zonal mean meridional wind velocity**

The zonal mean meridional wind velocity $\overline{\langle v(z,\phi) \rangle}$ which also accounts for convective contribution is calculated by

$$\overline{\langle v(z,\phi) \rangle}$$
$$= \frac{d1*\left(-2\tan(\phi)\left(\overline{\langle u^*v^* \rangle} + \overline{\langle u'v' \rangle}\right)\right) + d2*\left(\frac{\partial}{\partial \phi}\left(\overline{\langle u^*v^* \rangle} + \overline{\langle u'v' \rangle}\right)\right) + d3*\left((-\frac{dK_z}{z}+\frac{K_z}{H_0})\frac{\partial \overline{\langle u \rangle}}{\partial z}a\right) + d4*(A)}{n1*(\tan(\phi)\overline{\langle u \rangle}) + n2*\left(-\frac{\partial \overline{\langle u \rangle}}{\partial \phi}\right) + n3*(2\Omega a \sin(\phi))}$$

$$( S7\,S9 )$$

With $K_z = 0.005\, z$ and In (4 )

$$A = \left(\frac{P_{conv}\, L}{(\Gamma_a - \Gamma)} - \frac{1}{H_0}\right)\langle u_{s\_profile} \rangle$$

Whereby Γ is the lapse rate in the troposphere calculated by using the formula from Petoukhov (Petoukhov et al., 2000) , $P_{conv}$ by the cloud module implemented by Eliseev at al. (Eliseev, n.d.) and

$$\langle u_{s\_profile}\rangle = \begin{cases} 2, & |\phi| > 40 \\ -2\cos\left(\phi\frac{\pi}{40°}\right), & \text{otherwise} \end{cases}$$

$$A = \frac{\mathcal{L}\langle\overline{P_{co}}\rangle}{H_0}\frac{\overline{\langle u_{sf}\rangle}}{\Gamma_a - \Gamma_0 - \Gamma_1(T_a - T_0)\left(1 - a_q q_s^2\right) + \Gamma_2 n_c}$$

whereby the parameters are given in **Table 1**. We roughly approximate $\overline{\langle u_{sf}\rangle}$ by constant profile for this experiment

$$\overline{\langle u_{sf}\rangle} = \begin{cases} 2, & |\phi| > 40 \\ -2\cos\left(\phi\frac{\pi}{40°}\right), & \text{otherwise} \end{cases} \qquad (S10)$$

The additional  calculation of $\overline{\langle u_{sf}\rangle}$ instead of  the calculated surface zonal velocity is done to avoid instabilities. Instabilities can emerge due to the strong positive feedback between the meridional temperature and vertical wind velocity, which lead to high latent heat. In nature these would be damped out but due to fixed troposphere height, we parameterize it in the above described way.

For the derivation we start with the differential equation of the zonal wind component

$$\frac{du}{dt} = \frac{\tan\phi}{a}uv + fv - \frac{1}{\rho}\Delta_\lambda p + F_u \qquad (\text{S9}\ \text{S11})$$

Whereby a is the Earth radius, $f$ is the Coriolis factor and $F_u$ is the frictional force in $u$-direction. Multiplying the equation with $\rho$ and using that $\rho\frac{du}{dt} = \frac{d(\rho u)}{dt} - u\frac{d\rho}{dt}, \frac{d(\rho u)}{dt} = \frac{\partial(\rho u)}{\partial t} + V\cdot\Delta\,(\rho u)$ and $V\cdot\Delta\,(\rho u) = \Delta\,(\rho u V) - (\rho u)\Delta\cdot V$, we get

$$\frac{\partial(\rho u)}{\partial t} + \Delta\,(\rho u V) - u\left(\frac{d\rho}{dt} + (\rho u)\Delta\cdot V\right) = \frac{\tan\phi}{a}\rho uv + f\rho v - \Delta_\lambda p + \rho F_u$$

With the continuity equation and using spherical coordinates, the equation simplifies to

$$\frac{\partial(\rho u)}{\partial t} + \frac{1}{a\cos\phi}\frac{\partial(\rho u^2)}{\partial\lambda} + \frac{1}{a\cos\phi}\frac{\partial(\rho\cos\phi\,uv)}{\partial\phi} + \frac{\partial(\rho wu)}{\partial z}$$
$$= \frac{\tan\phi}{a}\rho uv + f\rho v - \frac{1}{a\cos\phi}\frac{\partial p}{\partial\lambda} + \rho F_u \qquad (S12)$$

$$\frac{\partial(\rho u)}{\partial t} + \frac{1}{a\cos\phi}\frac{\partial(\rho u^2)}{\partial\lambda} + \frac{1}{a\cos\phi}\frac{\partial(\rho\cos\phi\,uv)}{\partial\phi} + \frac{\partial(\rho wu)}{\partial z}$$
$$= \frac{\tan\phi}{a}\rho uv + f\rho v - \frac{1}{a\cos\phi}\frac{\partial p}{\partial\lambda} + \rho F_u \qquad (S10)$$

We calculate the zonal average $(\overline{\dots})$, take into account that $\frac{\overline{\partial x}}{\partial\lambda} = 0$ and assume a vertical dependence

of the density $\left(\rho = \rho_0(z)\right)\left(\rho = \rho_0(z)\right)$:

$$\frac{\overline{\partial(\rho_0 u)}}{\partial t} + \frac{1}{a}\frac{\overline{\partial(\rho_0 uv)}}{\partial\phi} + \frac{\overline{\partial(\rho_0 wu)}}{\partial z} = 2\frac{\tan\phi}{a}\rho\overline{uv} + f\rho\bar{v} + \rho_0\overline{F_u}$$

We split the wind variables into an synoptic scale waves, planetary waves and zonal mean wind ($u = \bar{u} + u^* + u'$). Under the assumption that $\bar{u}$ and $v^*$ are independent, the result of the zonal mean over the azonal component is zero:

$$\overline{uv} = \overline{\bar{u}\bar{v} + \bar{u}v^* + \bar{u}v' + u^*\bar{v} + u^*v^* + u^*v' + u'\bar{v} + u'v^* + u'v'}$$
$$= \overline{\bar{u}\bar{v} + \bar{u}v' + u^*v^* + u'\bar{v} + u'v'}$$

We average eq. (10) over time and phase speed ($\langle\dots\rangle$). By assuming independency of the variables, we can simplify the terms $(\overline{u'v'}) = $ (S12) over time and phase speed ($\langle\dots\rangle$). Due to a "gap" in the three-dimensional (period-wavelength-phase velocity) spectrum of atmospheric processes (see, e.g., Fraedrich & Böttger 1978, Coumou et al. 2011), the synoptic-scale component in its interaction with the large-scale long-term component of the atmospheric fields on the time scales about 10-20 days and longer could be, to a first approximation, represented (described) in terms of its ensemble (statistical) characteristics (the second and higher-order moments), and not as the individual eddies (Saltzman, 1978). We can simplify the terms $\langle\overline{u'v'}\rangle = \langle u'\bar{v}\rangle = 0$. In addition, it is $\overline{\bar{u}\bar{v}} = \bar{u}\bar{v}$ due to quasi stationarity of both terms. It is also $\langle\frac{dx}{dt}\rangle = 0$ and $\langle\bar{u}\bar{v}\rangle = \langle\bar{u}\rangle\langle\bar{v}\rangle$ since the oscillations of $\bar{u}$ and $\bar{v}$ are very small and independent of each other. By using the continuity equation $\frac{\rho_0}{a}\frac{\partial\langle\bar{v}\rangle}{\partial\phi} - \frac{\tan\phi}{a}\rho_0\bar{v} + \frac{\partial(\rho_0\langle\bar{w}\rangle)}{\partial z} = 0$, we can simplify eq. (S10 S12) to

$$\frac{1}{a}\rho_0\langle\bar{v}\rangle\frac{\partial\langle\bar{u}\rangle}{\partial\phi} + \frac{\rho_0}{a}\frac{\partial(\langle\overline{v^*u^*}\rangle + \langle\overline{v'u'}\rangle)}{\partial\phi} + \rho_0\langle\bar{w}\rangle\frac{\partial\langle\bar{u}\rangle}{\partial z} + \frac{\partial(\rho_0\langle\overline{w^*u^*}\rangle + \langle\overline{w'u'}\rangle)}{\partial z}$$
$$= \frac{\tan\phi}{a}\rho_0\langle\bar{u}\rangle\langle\bar{v}\rangle + 2\frac{\tan\phi}{a}(\langle\overline{v^*u^*}\rangle + \langle\overline{v'u'}\rangle) + f\rho\bar{v} + \rho_0\overline{F_u} \qquad (S11 \; S13)$$

With the assumption that $\rho_0 = e^{-z/H_0}$ and $-\rho_0 \bar{F}_u = \frac{\partial \bar{\tau}}{\partial z} = \frac{\partial}{\partial z}\left(\kappa \rho_0 \frac{\partial \langle \bar{u}\rangle}{\partial z}\right) = \kappa \frac{\partial \rho_0}{\partial z}\frac{\partial \langle \bar{u}\rangle}{\partial z} + \rho_0 \frac{\partial \kappa}{\partial z}\frac{\partial \langle \bar{u}\rangle}{\partial z} + \rho_0 \kappa \frac{\partial^2 \langle \bar{u}\rangle}{\partial z^2} = -\kappa \frac{\rho_0}{H_0}\frac{\partial \langle \bar{u}\rangle}{\partial z} + \rho_0 \frac{\partial \kappa}{\partial z}\frac{\partial \langle \bar{u}\rangle}{\partial z}$, we obtain

$$\rho_0 \langle \bar{v}\rangle \left(\frac{1}{a}\frac{\partial \langle \bar{u}\rangle}{\partial \phi} - \frac{\tan\phi}{a}\langle \bar{u}\rangle - f\right) = 2\frac{\tan\phi}{a}\left(\langle v^* u^*\rangle + \overline{\langle v'u'\rangle}\right) - \frac{\rho_0}{a}\frac{\partial\left(\overline{\langle v^* u^*\rangle} + \overline{\langle v'u'\rangle}\right)}{\partial\phi} - \rho_0\langle \bar{w}\rangle\frac{\partial\langle\bar{u}\rangle}{\partial z} -$$

$$\frac{\partial(\rho_0\langle \overline{w^* u^*}\rangle + \overline{\langle w'u'\rangle})}{\partial z} - \kappa\frac{\rho_0}{H_0}\frac{\partial\langle\bar{u}\rangle}{\partial z} + \rho_0\frac{\partial\kappa}{\partial z}\frac{\partial\langle\bar{u}\rangle}{\partial z} \qquad (\text{S12 S14})$$

The contribution to the vertical exchange of the atmospheric momentum from stationary eddies described in our case by zonally averaged $\langle \overline{w^* u^*}\rangle$ is shown negligibly small (Hantel and Hacker, 1978).  Also, the scale analysis attests that $\langle\bar{w}\rangle\frac{\partial\langle u\rangle}{\partial z}$ are small (Petoukhov et al., 2003);

$$-\rho_0\langle\bar{w}\rangle\frac{\partial\langle\bar{u}\rangle}{\partial z} - \frac{\partial(\rho_0\langle\overline{w^* u^*}\rangle + \overline{\langle w'u'\rangle})}{\partial z} - \kappa\frac{\rho_0}{H_0}\frac{\partial\langle\bar{u}\rangle}{\partial z} + \rho_0\frac{\partial\kappa}{\partial z}\frac{\partial\langle\bar{u}\rangle}{\partial z} \approx -\frac{\partial(\rho_0\overline{\langle w'u'\rangle})}{\partial z} - \kappa\frac{\rho_0}{H_0}\frac{\partial\langle\bar{u}\rangle}{\partial z} + \rho_0\frac{\partial\kappa}{\partial z}\frac{\partial\langle\bar{u}\rangle}{\partial z}$$

Hence the eq. ( S14) can be rewritten into

$$\rho_0 \langle \bar{v}\rangle \left(\frac{1}{a}\frac{\partial \langle \bar{u}\rangle}{\partial \phi} - \frac{\tan\phi}{a}\langle \bar{u}\rangle - f\right) = 2\frac{\tan\phi}{a}\left(\langle v^* u^*\rangle + \overline{\langle v'u'\rangle}\right) - \frac{\rho_0}{a}\frac{\partial\left(\overline{\langle v^* u^*\rangle} + \overline{\langle v'u'\rangle}\right)}{\partial\phi} - \rho_0\langle \bar{w}\rangle\frac{\partial\langle\bar{u}\rangle}{\partial z} -$$

$$\frac{\partial(\rho_0\langle \overline{w^* u^*}\rangle + \overline{\langle w'u'\rangle})}{\partial z} - \kappa\frac{\rho_0}{H_0}\frac{\partial\langle\bar{u}\rangle}{\partial z} + \rho_0\frac{\partial\kappa}{\partial z}\frac{\partial\langle\bar{u}\rangle}{\partial z} \qquad (\text{S13 S15})$$

With $\langle\overline{u'w'}\rangle = -\kappa'\frac{\partial\langle\bar{u}\rangle}{\partial z}$, whereby $\kappa'$ is the coefficient of large-scale turbulent exchange for the momentum due to transient synoptic eddies  (Williams and Davies, 1965), we get

$$\rho_0\langle\bar{v}\rangle\left(\frac{1}{a}\frac{\partial\langle\bar{u}\rangle}{\partial\phi} - \frac{\tan\phi}{a}\langle\bar{u}\rangle - f\right)$$

$$= 2\frac{\tan\phi}{a}\left(\langle v^* u^*\rangle + \overline{\langle v'u'\rangle}\right) - \frac{\rho_0}{a}\frac{\partial\left(\overline{\langle v^* u^*\rangle} + \overline{\langle v'u'\rangle}\right)}{\partial\phi} - (\kappa+\kappa')\frac{\rho_0}{H_0}\frac{\partial\langle\bar{u}\rangle}{\partial z} + \rho_0\frac{\partial(\kappa+\kappa')}{\partial z}\frac{\partial\langle\bar{u}\rangle}{\partial z}$$

With $\tilde{\kappa} = \kappa + \kappa'$ $K_z = \kappa + \kappa'$ we can simplify the equation to

$$\overline{\langle v(z,\phi)\rangle} = \frac{-2\tan(\phi)\left(\overline{\langle u^* v^*\rangle} + \overline{\langle u'v'\rangle}\right) + \frac{\partial}{\partial\phi}\left(\overline{\langle u^* v^*\rangle} + \overline{\langle u'v'\rangle}\right) + \left(-\frac{dK_z}{dz} + \frac{K_z}{H_0}\right)\frac{\partial\overline{\langle u\rangle}}{\partial z}a}{\tan(\phi)\overline{\langle u\rangle} - \frac{\partial\overline{\langle u\rangle}}{\partial\phi} + 2\Omega a\sin(\phi)}$$

The derived equation for the meridional velocity does not account for latent heat release associated with convective precipitation. To capture this additional term we include convective precipitation and finally introduce tuning parameters, which have values close to 1.

**S3 Schematic plot of the optimization process**